# Effect of scour on the fatigue life of offshore wind turbines and its prevention through passive structural control

Yu Cao[1], Ningyu Wu[2], Jigang Yang[2], Chao Chen[1,3*], Ronghua Zhu[3,4*], Xugang Hua[1],

[1] Key Laboratory for Bridge and Wind Engineering of Hunan Province, College of Civil Engineering, Hunan University, Changsha, China

[2] Hebei Construction Investment Offshore Wind Power Co., Ltd., Tangshan, China

[3] Yangjiang Offshore Wind Laboratory, Yangjiang, China

[4] Ocean College, Zhejiang University, Hangzhou, China

*Corresponding author: steinchen@hnu.edu.cn, zhu.richard@zju.edu.cn

**Abstract**
Offshore wind turbine (OWT) support structures are exposed to the risk of fatigue
damage and scour, and this risk can be effectively mitigated by installing structural
control devices such as tuned mass dampers (TMDs). However, time-varying scour al-
tering OWTs' dynamic characteristics has an impact on the TMD design and fatigue
life, which was rarely studied before. In this paper, a simplified modal model is used to
investigate the influence of scour and a TMD on the fatigue life evaluation of a 5 MW
OWT's support structure, and a traditional method and a newly developed optimization
technique are both presented to obtain TMD parameters. This optimization technique
aims at finding optimal parameters of the TMD which maximizes the fatigue life of a
hotspot at the mudline, and the effect of time-varying scour can be considered. This
study assumes that the TMD operates in the fore-aft (FA) direction, while the vibration
in the side-side (SS) direction is uncontrolled. Results show that scour can decrease the
fatigue life by about 24.1 %, and the TMD can effectively suppress vibration and in-
crease the fatigue life. When the scour depth reaches 1.3 times the pile diameter, the
TMD with a mass ratio of 1 % can increase the fatigue life of OWT's support structure
by about 64.6 %. Further, it is found that the fatigue life can be extended by 25 % with
the TMD optimized by the proposed optimization technique, compared to that with the
traditionally design method, which does not take the change of dynamic characteristics
into account.
**Keywords:** scour, offshore wind turbine, structural control, modal analysis, fatigue life.

1  **Introduction**
With the continuous development of large-size fixed-bottom OWTs, local scour
and scour protection of pile foundation have become a common issue (L. Wang et al.,
2020; X. Wang et al., 2019; F. Zhang et al., 2022). Scour has a significant impact on
the dynamic characteristics, vibration magnitudes, and thus fatigue life of OWTs under
wind and wave loads. On the one hand, the action of currents and waves causes local
scour pits around pile foundations, which reduces the burial depth of pile foundations.
This phenomenon usually causes a reduction in the natural frequencies of OWTs and
changes in other dynamic characteristics. This can potentially lead to resonance, large
amplitude stress cycles and fatigue damage when one of the natural frequencies is close
to the rotational frequency of the blades (Sørensen and Ibsen, 2013). On the other hand,
current scour protection measures cannot completely avoid scour and have their own
shortcomings. For example, armouring protection has the disadvantage that the projec-
tile cannot be accurately cast in complex sea conditions and is easy to be washed away
(G. Wang et al., 2023; F. Zhang et al., 2023). Flow-altering protection has the disad-
vantages of high cost and changing the dynamic characteristics of the foundation (Tang
et al., 2023). As offshore structures, wind turbines are vulnerable to corrosion from
seawater, which makes the fatigue problem worse (Amirafshari et al., 2021). Thus, the
scour-induced changes in dynamic characteristics and risk in resonance inevitably in-
duce a further increase in fatigue damage and deserve in depth research (Mayall et al.,

58  2018).

Many researchers have studied the effect of scour on fatigue damage accumulation
in OWTs. For instance, Tempel et al. (2006) investigated the frequency and fatigue of
piles under different scour depths and concluded that scour has a little effect on the
natural frequencies but a great effect on fatigue damage. Zhang et al. (2021) found that
scour depth has a significant influence on monopile impedance. Rezaei et al. (2018)
showed that scour leads to an increase in the maximum bending moment of the mono-
pile and a shortening of the fatigue life. To mitigate the fatigue damage in OWTs, in-
stalling structural control devices is an effective way. It was  demonstrated that TMDs
have a positive effect on reducing vibration amplitudes of wind turbine systems (Lack-
ner and Rotea, 2011a; Dinh and Basu, 2015; Lu et al., 2023; Aydin et al., 2023). Dai et
al. (2021) conducted a shaker experiment using a scaled wind turbine model and
showed that the installed TMD can suppress the vibration of the structure more effec-
tively considering soil-structure interaction (SSI).
In the previously mentioned studies, researchers have individually investigated the
effect of scour on structural vibration and fatigue, and the structural control by TMDs
for OWTs. However, in practice, the effect of scour combining structural control via
TMDs could have a significant impact on OWTs' fatigue life. Moreover, whether con-
sidering scour could influence the design of TMDs, and TMDs with different parame-
ters can also have an impact on fatigue damage accumulation.
The purpose of this study is to explore the effect of scour on the fatigue life of
wind turbine structures and the control effect of TMD on the fatigue life of wind turbine
structures under scour conditions. The authors present a case study with a 5 MW single-
pile wind turbine to carry out related research. In this study, ABAQUS is used to estab-
lish a detailed SSI model with different scour depths. A finite element (FE) model con-
sidering wind loads and TMD is established in MATLAB, and the scour effect is con-
sidered by establishing a relationship with the ABAQUS model by means of the
equivalent stiffness matrix. And the finite element model is simplified to a modal model
for fast prediction of fatigue life. The TMD operates in the FA direction and does not
work in the SS direction. This study investigates the effect of different scour depths on
the performance of the TMD and the fatigue life of a 5 MW OWT's support structure
including a tower and a monopile foundation, and the optimization of the TMD's pa-
rameters considering time-varying scour depths to maximum fatigue life is also pre-
sented. This study provides some knowledge of the effects of the time varying scour
and the TMD on the fatigue life of wind turbines, as well as a new TMD design method
targeting at enhancing fatigue resistance. The rest of the paper is organized as follows:
Section 2 introduces the numerical models used in the research. Section 3 introduces
the traditional TMD design method and the newly developed parameter optimization
method. Section 4 describes the load cases for the fatigue analysis, the analysis results
of this study and the TMD parameter optimization results. Section 5 concludes the
study.
## 2  Model description
### 2.1  Finite element model and implementation of tuned mass damper
An FE model of a monopile-supported OWT installed with a TMD is established
in MATLAB. This model contains a flexible tower, a rotor-nacelle assembly (RNA),
and an external TMD, considering the foundation flexibility. The model is based on the
widely used NREL 5MW reference OWT, and its detailed properties are shown in Ta-
ble 1. Three-dimensional beam elements are used to create the FE model and the theo-
retical basis is the standard Euler-Bernoulli beam theory. The wind turbine tower is
divided into 18 beam elements, and the monopile between the mudline and the mean
sea level (MSL) are divided into 4 beam elements. A convergence test by comparing
the first natural frequencies shows that 22 beam elements are sufficient. Each element
node has 6 degrees of freedom (DOFs) corresponding to translational and rotational
motions in various directions. The mass matrix and stiffness matrix in the equation of
motion of the OWT structure can be obtained using the material properties. The damp-
ing matrix is applied by means of Rayleigh damping, and the combined damping ratio
of soil damping and structural damping is assumed to be 1 % (Chen and Duffour, 2018).
The Rayleigh mass and stiffness coefficients $\alpha_1$ and $\alpha_2$ are defined by $\alpha_1 = \alpha_2 =$
$\frac{\zeta_C}{\frac{1}{2\omega}+\frac{\omega}{2}}$. $\omega$ is the natural frequency of the first FA mode, and $\zeta_C$ is the combined damping
ratio. The RNA is represented by a lumped mass at the tower top.

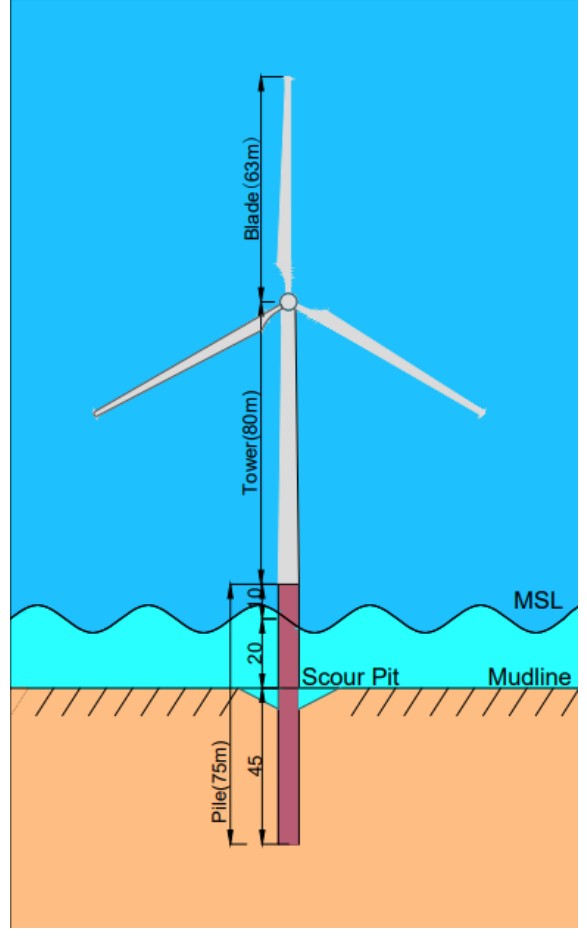


Fig. 1. Schematic of NREL 5MW wind turbine and scour effect
The TMD is mounted on the top of the tower, and the effect of the TMD is consid-
ered by incorporating its mass, damping, and stiffness terms at relevant positions in the
local mass, damping, and stiffness matrices of the beam element representing the top
of the tower. The equation of motion of the OWT main structure is:

$$\mathbf{M}_s\ddot{\mathbf{U}}_s + \mathbf{C}_s\dot{\mathbf{U}}_s + \mathbf{K}_s\mathbf{U}_s + \mathbf{C}_T(\dot{\mathbf{U}}_s - \dot{\mathbf{U}}_T) + \mathbf{K}_T(\mathbf{U}_s - \mathbf{U}_T)$$
$$= \mathbf{F}_{\text{wind}} + \mathbf{F}_{\text{wave}},$$

(1)

where $\mathbf{M}_s$, $\mathbf{C}_s$, $\mathbf{K}_s$ are the mass, damping and stiffness matrices of the main structure.

$\mathbf{C}_T$, $\mathbf{K}_T$ are matrices with same dimensions containing $c_T$, $k_T$, $\mathbf{C}_T = \begin{bmatrix} 0 & \cdots & 0 \\ \vdots & \ddots & \vdots \\ 0 & \cdots & c_T \end{bmatrix}$, $\mathbf{K}_T =$

$\begin{bmatrix} 0 & \cdots & 0 \\ \vdots & \ddots & \vdots \\ 0 & \cdots & k_T \end{bmatrix}$. $\mathbf{U}_s$ is the displacement vector of the main structure, $\mathbf{U}_s = \begin{bmatrix} u_{s-1} \\ \vdots \\ u_{s-top} \end{bmatrix}$.

$\mathbf{U}_T$ is the displacement vector containing $u_T$, $\mathbf{U}_T = \begin{bmatrix} 0 \\ \vdots \\ u_T \end{bmatrix}$. $\mathbf{F}_{wind}$, $\mathbf{F}_{wave}$ are the aerody-

namic and wave load vectors. The equation of motion for the TMD can be represented
by

$$m_T \ddot{u}_T + c_T(\dot{u}_T - \dot{u}_{s-top}) + k_T(u_T - u_{s-top}) = 0, \tag{2}$$

where $m_T$, $c_T$, $k_T$ are the mass, damping and stiffness of the TMD, $u_T$, $u_{s-top}$ are the
displacement of the TMD and the displacement of the top node. The modelling of SSI
is realized by an equivalent stiffness matrix, which will be introduced in detail subse-
quently in Section 2.3.

Table 1. Basic properties of the NREL 5MW reference OWT (J. Jonkman et al., 2009;
Rezaei, 2017)

| | |
|---|---|
| Number of blade | 3 |
| Rotor diameter | 126 m |
| Tower length | 80 m |
| Tower diameter | 3.87–6.00 m |
| Tower thickness | 28–38 mm |
| Pile length | 75 m |
| Pile penetration depth | 45 m |
| Pile diameter | 6 m |
| Pile thickness | 80 mm |
| Hub height from MSL | 92.4 m |
| Turbine mass | 350000 kg |
| Blade mass | 17740 kg |
| Rated wind speed | 12.1 m/s |

Wind loads were calculated using modified unsteady blade element momentum
(BEM) theory (Branlard, 2017; B. J. Jonkman and Buhl, 2006) with Prandtl and Glauert
corrections. Ignoring the iterative loop (Chen, Duffour, Fromme, et al., 2021) in the
steady-state BEM code, the instantaneous aerodynamic forces were calculated for each

time step within the time integration. The turbulent wind field was generated using the
Kaimal spectrum according to the wind field parameters of IEC 61400-3 (2019) as-
suming moderate turbulence intensity. It should be noted that the aerodynamic loads
from the rotor applied at the tower top were calculated using an aerodynamic force
linearization technique previously developed by the authors (Chen, Duffour, Fromme,
et al., 2021; Chen et al., 2020). This technique divides the aerodynamic loads into two
parts. The first part is the quasi-steady aerodynamic force calculated by BEM theory,
which does not consider the influence of tower top motion. The second part considers
the effect of aerodynamic damping by introducing an additional aerodynamic damping
matrix. The adoption of this technique aims to facilitate the development of a simplified
modal model for rapid fatigue calculation, which will be introduced in detail in Sub-
section 2.4. To represent the influence of controller in the OWT, a standard relationship
(J. Jonkman et al., 2009) between the mean wind speed, rotor rotation speed, and blade
pitch angles, which represents the OWT's normal operational conditions, are adopted
throughout the wind loading calculations.

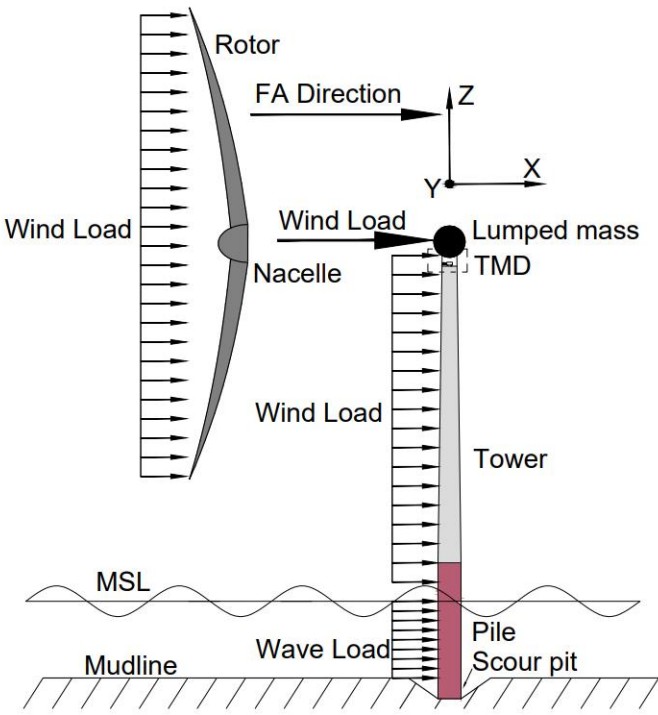


Fig. 2 Schematic of wind turbine load application

Wave loads were calculated using the Morison equation, which includes viscous

drag and inertial forces:

$$\mathbf{F}_{\text{wave}} = \frac{1}{2}\rho_w D_{\text{pile}} C_d |\dot{\mathbf{u}}_w| \dot{\mathbf{u}}_w + \frac{\pi}{4}\rho_w D_{\text{pile}}{}^2 C_m \ddot{\mathbf{u}}_w, \qquad (3)$$

where $\dot{\mathbf{u}}_w$ and $\ddot{\mathbf{u}}_w$ are the velocity and acceleration of water particles, $C_d$ is the drag coefficient, $D_{\text{pile}}$ is the diameter of the monopile between the mean sea level and the mudline, $C_m$ is the inertia coefficient and $\rho_w$ is the density of water. $C_d$ and $C_m$ were chosen as 1 and 2 respectively as the recommended values in Shirzadeh et al (2013). The wave profiles were obtained through the superposition of wave components, combining linear wave theory and JONSWAP spectra (Klaus et al., 1973). The application of wind and wave loads is shown in Fig. 2.

## 2.2 Scour modelling in ABAQUS

Using solid elements to model pile-soil interaction (S. Dai et al., 2021; Fard et al., 2022; Ma and Chen, 2021; Zdravković et al., 2015) is usually considered more accurate than the p-y curve method (Liang et al., 2018; Song and Achmus, 2023) and the equivalent embedding method (Shahmohammadi and Shabakhty, 2020; Bergua et al., 2022). The solid element method can also reduce the influence of empirical formula on the results. Therefore, the solid element method is used to establish the wind turbine scour model. The wind turbine scour model established in ABAQUS contains soil, pile foundation, tower, and the RNA is replaced by a concentrated mass located at the top of the tower. The diameter of the soil body is selected as 20 times the pile diameter, the soil under the pile foundation is selected as 2.5 times the pile diameter, and the total height of the soil body is 60 m. The soil body is made of homogeneous dense sandy soil, and the pile and tower are made of steel. The material parameters of the soil body, pile and tower are shown in Table 2 below:

Table 2. Soil, pile and tower material parameters

| Type | Weight $\gamma$ (kN/m³) | modulus of elasticity $E_S$ (MPa) | Poisson's ratio $\upsilon$ | Internal friction angle $\varphi$ (°) | Expansion angle $\psi$ (°) | Cohesion c (kPa) |
|------|------|------|------|------|------|------|
| Soil | 19 | 80 | 0.3 | 35 | 23 | 0.1 |
| Pile | 78.5 | 215 | 0.25 | - | - | - |
| Tower | 85 | 215 | 0.25 | - | - | - |

The Mohr-Coulomb model is used for the soil, and the pile, tower, and nacelle are assumed to be elastic since they are much stiffer than the soil and do not deform

plastically under normal operational conditions. The pile and tower are connected by a
binding relationship. The normal contact between the pile and soil adopts the hard con-
tact, and the tangential contact adopts the friction penalty function. The relative sliding
friction factor at the interface, $\mu$ is equal to $\tan(0.75\,\varphi)$, where $\varphi$ is the internal friction
angle. The pile-soil contact is in the form of frictional contact, where mutual contact
pairs are established between the pile and the soil, including the contacts between the
pile bottom surface and the soil, the outside surface of the pile and the soil, and the
inside surface of the pile and the soil core. The frictional contact between pile bottom
surface and soil is omitted due to the small area of the contact surface. These frictional
contacts all adopt the face-to-face contact, and the contact discretization method adopts
the face-to-face discretization method, considering the large stiffness of the main sur-
face and the small stiffness of the slave surface. The perimeter of the soil body is trans-
lationally constrained, and the bottom surface of the soil adopts a fixed constraint. The
eight-node linear brick element (C3D8R) is used to model the pile and soil, and the
mesh division is realized by arranging seeds as shown in Fig. 3. The whole model is set
up by adopting the modelling method of "element birth and death", which realizes the
operation of initial soil stress balance operation and sets up contacts and other related
steps by killing and activating relevant elements.

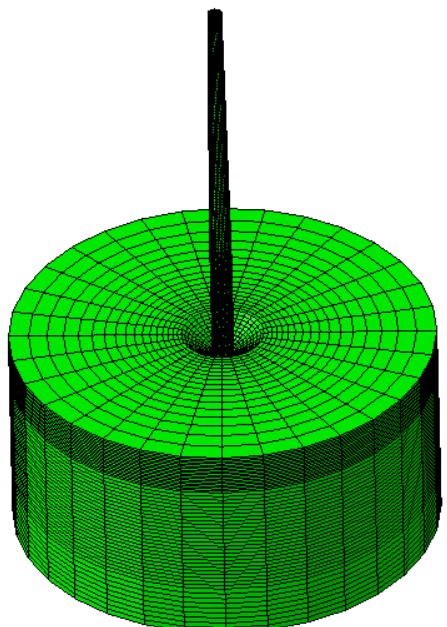


Fig. 3. Pile-soil interaction modelled by ABAQUS
The scour conditions can be represented by a deep conical pit around the pile under
the long-term action of the waves and currents. According to the specification of Det
Norske Veritas (DNV) (2014b), the radius of the pit surface formed by scour, R, can
be related to the depth of the scour pit by

$$R = \frac{D}{2} + \frac{S}{\tan\varphi}, \tag{4}$$

where D is the diameter of the pile, S is the scour depth, and $\varphi$ is the angle of internal
friction of the soil.
2.3 **Equivalent stiffness matrix method**
It is necessary to consider the effect of scour in the FE model in MATLAB. An
equivalent stiffness matrix method is adopted in the FE model to consider the flexibility
induced by SSI. The 6 DOFs of node at the mudline are assumed to be constrained by
a series of coupled springs, and the stiffnesses of the coupled springs form a 6×6 stiff-
ness matrix. For a specific stiffness term used in the FE model, for instance the one
relevant to the lateral displacement in the FA direction, the value of the stiffness term
can be found from the relationship between the reaction force at the mudline and the
pile top displacement (Jung et al., 2015). The equivalent stiffness schematic of the pile-
soil interaction in the FA direction for the OWT is shown in Fig. 4.
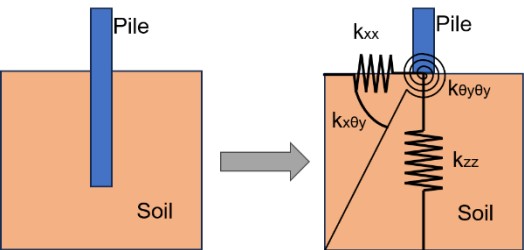
Fig. 4. Equivalent stiffness schematic of pile-soil interaction in the FA direction
According to the principle of virtual displacement and with the DOFs in other
directions constrained, a unit displacement or rotation is first applied in one direction,
and then the reaction force in that direction can be determined. The equivalent stiffness
in that direction can be subsequently calculated by the relationship between the dis-
placement and reaction force. Using the same approach, the stiffness terms correspond-
ing to the remaining five DOFs are calculated. The stiffness terms in all the 6 DOFs
together form all the diagonal terms of the soil stiffness matrix. With the diagonal terms
known, the off-diagonal stiffness terms can be found by applying a unit displacement
in one direction and looking at the reaction force in the other concerned direction, with
the other four DOFs constrained. Using the same principle, the off-diagonal terms can
also be found from the relationship between the displacements and reaction forces,
which ultimately results in a 6×6 stiffness matrix (Bergua et al., 2021; Pedersen and
Askheim, 2021):

$$\mathbf{F_{soil}} = \begin{Bmatrix} F_x(t) \\ F_y(t) \\ F_z(t) \\ M_x(t) \\ M_y(t) \\ M_z(t) \end{Bmatrix} = \begin{bmatrix} k_{xx} & 0 & 0 & 0 & k_{x\theta y} & 0 \\ 0 & k_{yy} & 0 & k_{y\theta x} & 0 & 0 \\ 0 & 0 & k_{zz} & 0 & 0 & 0 \\ 0 & k_{\theta xy} & 0 & k_{\theta x\theta x} & 0 & 0 \\ k_{\theta yx} & 0 & 0 & 0 & k_{\theta y\theta y} & 0 \\ 0 & 0 & 0 & 0 & 0 & k_{\theta z\theta z} \end{bmatrix} \begin{Bmatrix} u_x(t) \\ u_y(t) \\ u_z(t) \\ \theta_x(t) \\ \theta_y(t) \\ \theta_z(t) \end{Bmatrix} \quad (5)$$

$$= \mathbf{K_{soil}}\mathbf{u_{soil}} \,,$$

where $\mathbf{K_{soil}}$ is the equivalent soil stiffness matrix, $\mathbf{u_{soil}}$ is the displacement vector, and
$\mathbf{F_{soil}}$ is the reaction force vector. The equivalent soil stiffness matrix ignores the nonlin-
earity in the force-displacement relationship. This approach is suitable for fatigue anal-
ysis, as in normal operation conditions the deformation of the soil around the monopile
is relatively small and the nonlinearity in soil stiffness is very weak. The 6×6 soil stiff-
ness matrix obtained from ABAQUS is imported to the FE model in MATLAB. This
modelling method combines the accuracy enhancement provided by the scour model in
ABAQUS using solid elements with the fast calculation speed and convenience of ap-
plying wind and wave loads using the FE model in MATLAB.
2.4 **Rapid fatigue evaluation method**
The established FE model in MATLAB can generate dynamic responses of the
OWT, considering wind and wave loads and scour effect. However, a comprehensive
fatigue life prediction in time domain needs to consider a large number of environmen-
tal states and load cases, so simulation efficiency is very important. Moreover, the TMD
design optimization requires much more dynamic response time series. The FE model
is not fast enough in this case. Therefore, a simplified modal model is developed from
the FE model in MATLAB following the method develop in Ref. (C .Chen et al., 2021).
The total aerodynamic forces from the rotor applied on the tower top node are linearized
as the sum of a term corresponding to the forces for an assumed rigid tower, and a term
proportional to the tower top linear and angular velocities. The hydrodynamic forces
are linearized by ignoring the relatively small monopile vibrations. The details for force
linearization can be found in the authors' previous studies (Chen, Duffour, Fromme, et
al., 2021). Since the dynamic responses of the OWT are mainly dominated by the first
two bending vibration modes, the FE model is reduced into a 4-DOF simplified modal
model by considering only the first two bending modes in the FA and SS directions
respectively. The development of the simplified 4-DOF modal model is briefly intro-
duced as follows. Denoting the mass matrix and stiffness matrix of the OWT as $\mathbf{M}$ and
$\mathbf{K}$ including the TMD and the lumped soil stiffness matrix, the undamped vibration
mode matrix $\mathbf{\Psi}$ can be obtained directly through eigen analysis. According to the rela-
tionship $\mathbf{u} = \mathbf{\Psi}\boldsymbol{\alpha}$ and multiplying the transpose of the undamped vibration matrix $\mathbf{\Psi^T}$
with the equation of motion, the following equation is obtained:

$$\mathbf{\Psi^T M \Psi \ddot{\alpha}} + \mathbf{\Psi^T C \Psi \dot{\alpha}} + \mathbf{\Psi^T K \Psi \alpha} = \mathbf{\Psi^T F}. \tag{6}$$

Then rewrite the above equation as

$$\mathbf{\bar{M}\ddot{\alpha}} + \mathbf{\bar{C}\dot{\alpha}} + \mathbf{\bar{K}\alpha} = \mathbf{\bar{F}}, \tag{7}$$

where $\boldsymbol{\alpha}$ is the general coordinate vector, $\mathbf{\bar{M}}$ is the modal mass matrix, $\mathbf{\bar{C}}$ is the modal
damping matrix, $\mathbf{\bar{K}}$ is the modal stiffness matrix, $\mathbf{\bar{F}}$ the modal load vector. Truncating
Eq. (6) by only considering the first two bending modes, the FE model is reduced to a
4-DOF modal model, which can be used for a rapid fatigue analysis. The dynamic re-
sponses of the OWT can be obtained by modal superposition after solving the general
coordinate vector by time integration. In the 4-DOF simplified modal model, the cross-
section stress at any height can be calculated from the calculated node displacements.
According to the dynamic stress extraction method provided by Pelayo et al. (Pelayo et
al., 2015), the cross-section stress $\sigma_Z(t)$ at any moment at a given location can be ob-
tained by:

$$\sigma_Z(t) = -E\big(\mathbf{N}^{e''}(z)\mathbf{u}_x^e(t)x + \mathbf{N}^{e''}(z)\mathbf{u}_y^e(t)y\big), \tag{8}$$

where $\mathbf{u}^e$ is the nodal displacement vector at the cross section, E is the material elastic
modulus, and $\mathbf{N}^e$ is the elemental shape function vector of FE model, x and y are the
positions within the section at the height z of the tower. After cyclic counting the stress
time series using the rainfall counting method, the fatigue damage at the hotspot can be
evaluated by utilizing the Palmgren-Miner rule based on the S-N fatigue calculation
method. The S-N curve for steel under water can be obtained by the following equation
considering the thickness effect in DNV (2014a):

$$\log N = \log \bar{a} - m \cdot \log \left[ \Delta \sigma \left( \frac{t}{t_{ref}} \right)^k \right], \tag{9}$$

where N is the number of cycles to failure, $\Delta\sigma$ is the stress range. $\Delta\sigma$ is calculated from the nominal stress $\Delta\sigma_{nominal}$ by the equation $\Delta\sigma = SCF \cdot \Delta\sigma_{nominal}$, SCF is the stress concentration factor. m is the negative inverse slope of the S-N curve, and $\log\bar{a}$ is the intercept between the log N axis and the S-N curve, $t_{ref}$ is the reference thickness for welded joints, t is the thickness at which cracks may grow. And $t = t_{ref}$ is used for thickness less than $t_{ref}$. When t is larger than $t_{ref}$, t is the actual thickness of the pile. k is the thickness exponent of fatigue strength. For pile joints, $t_{ref} = 25mm$. According to the DNV code, a bilinear S-N curve is usually used for offshore structures subjected mainly to typical wind and wave loads, using the Class E structural detail S-N curve shown in Table 3.

Table 3 Class E structural detail S-N curves

| $N \leq 10^6$ | | $N \geq 10^6$ | | | | |
|---|---|---|---|---|---|---|
| $m_1$ | $\log\bar{a}_1$ | $m_2$ | $\log\bar{a}_2$ | k | t(mm) | SCF |
| 3.0 | 11.610 | 5.0 | 15.350 | 0.2 | 80 | 1.13 |

For stresses with variable amplitudes, the fatigue damage index is calculated using the Palmgren-Miner summation rule:

$$D_k = \sum_{i=1}^{N_c} \frac{n_i}{N_i}, \tag{10}$$

where $N_c$ is the total number of bins, $n_i$ is the number of cycles in $i^{th}$ stress bin, $N_i$ is the number of cycles to failure for the $i^{th}$ stress range, and $D_k$ is the total fatigue damage index. The "rainflow" function in MATLAB is used for rainflow counting. When a stress time history is provided, this function can automatically calculate the $i^{th}$ stress range and the corresponding cycle number $n_i$, and $N_c$ is the total number of stress ranges. Fatigue failure occurs at the hotspot when the fatigue damage index reaches unit 1.

## 3 Damper design and optimisation method

Installing damping devices can efficiently reduce the vibration amplitudes of OWTs so that their service life can be greatly prolonged. Using TMDs as passive

control devices is the most widely used method to control the vibration of OWT support
structures. Usually, most TMDs are designed based on the dynamic characteristics of
the OWTs determined in the preliminary design stage, without considering the changes
in dynamic properties that may be caused by scour and soil degradation. In the real
environment, scour can alter the dynamic characteristics of OWTs, potentially reducing
the effectiveness of installed dampers or rendering them completely ineffective. There-
fore, it is of great significance to consider the changes in dynamic properties caused by
scour on the TMD design. The following two subsections first introduce the traditional
TMD design method considering constant dynamic characteristics in the initial state.
Then, an optimal parameter search method for the design of TMDs is presented, taking
into account the effects of scour and fatigue life evaluation.
## 3.1 TMD design in initial state

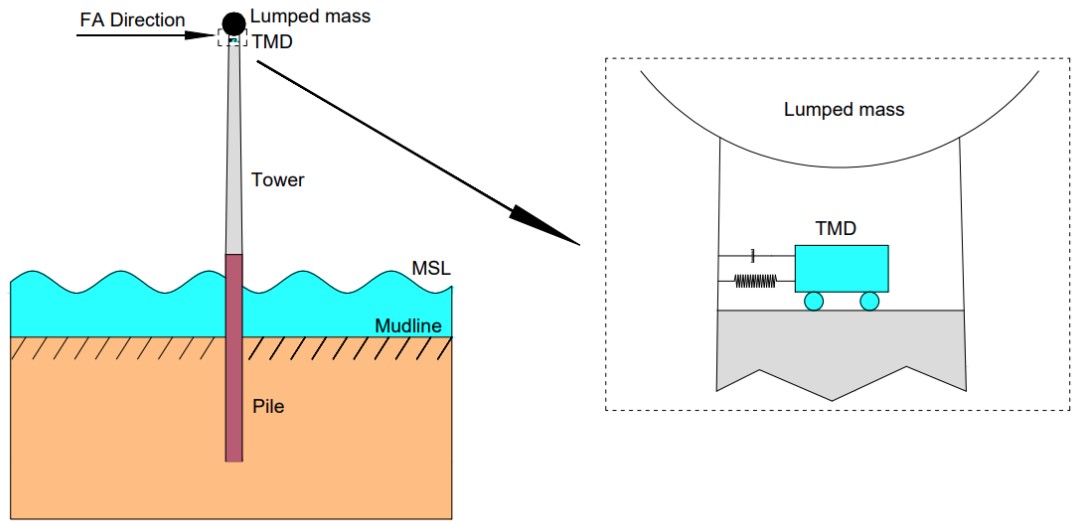


Fig. 5. Schematic diagram of TMD arrangement in the tower tube
As the dominant vibration mode of the OWT structure in operation is the first
bending mode, the largest vibration amplitude occurs at the top of the tower, making
the installation of the TMD at the tower top most effective. Therefore, the TMD is
installed inside the steel tube at the tower top to mainly control the vibration in the FA
direction, as shown in Fig. 5. And the TMD can be aligned with the FA direction by
rotating the damper. Accordingly, the initial design of the TMD is mainly carried out
based on the dynamic properties for the first-order mode. The initial design is conducted
based on the assumption that the monopile foundation is not scoured.
Numerous studies have shown that a TMD can effectively suppress the vibration
of a main structure when the mass ratio of the TMD to the main structure is 1 %-2 %
(Lackner and Rotea, 2011b; R. Zhang et al., 2019). After determining the mass ratio,
according to  the classic TMD optimization theory proposed by Den Hartog (1957), the
optimal frequency ratio of the TMD to the OWT structure is

$$\alpha_{opt} = \frac{1}{1 + \mu}. \tag{11}$$

The optimal damping ratio for the TMD can be calculated by

$$\xi_{opt} = \sqrt{\frac{3\mu}{8(1 + \mu)}}, \tag{12}$$

where $\mu$ is the mass ratio of the TMD to the OWT structure, $\alpha_{opt}$ is the optimal fre-
quency ratio of the TMD to the OWT structure and $\xi_{opt}$ is the optimal damping ratio of
the TMD.
Considering that excessive mass will lead to increased construction costs, difficul-
ties, and changes in the inherent characteristics of the original structure, the mass ratio
of the TMD system to the main structure is initially set at 1 %. Moreover, previous
studies have found that a TLCD with a mass ratio of 1 % and a TMD with a mass ratio
of 2 % can effectively suppress vibration (Colwell and Basu, 2009; Lackner and Rotea,
2011b; R. Zhang et al., 2019). According to Den Hartog's optimization theory for the
initial TMD design, it can be determined that the optimal frequency ratio of the TMD
to the main structure is 0.99, and the optimal damping ratio of the TMD is 0.061. When
the OWT support structure is not scoured, the first-order modal mass of the structure is
440350 kg, and the first-order modal frequency is 0.265 Hz. Therefore, according to
the initial design parameters, the mass, stiffness coefficient and damping coefficient of
the TMD system are 4403.5 kg, 11,952 N/m and 885 N s/m respectively.
3.2 **Fatigue-based damper optimisation technique**
After scouring occurs around the monopile foundation, the burial depth of the
monopile and natural frequencies of the OWT gradually change. The vibration mitiga-
tion effect of the TMD designed based on the dynamic parameters in the initial state
can be reduced, which may lead to an increase in fatigue damage to the OWT support
structure. Therefore, when designing the TMD, considering the influence of time-var-
ying scour can enhance the performance of the TMD and result in a longer fatigue life
of the support structure.

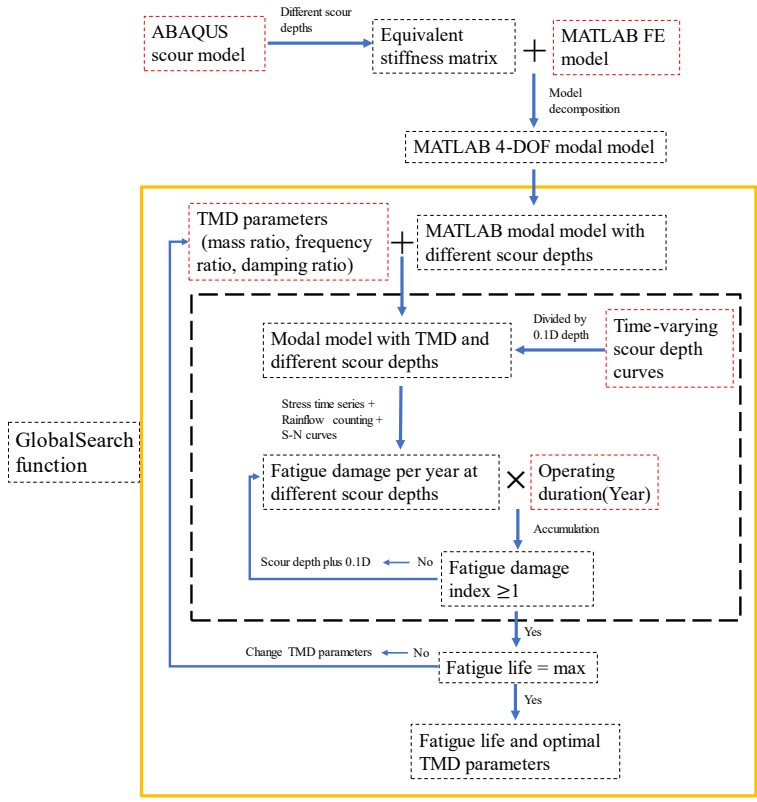


Fig. 6. Flowchart of TMD fatigue-life-based optimization technique

Here a fatigue-life-based optimization technique (FOT) to find optimal parameters

of the TMD is developed in MATLAB as shown in Fig. 6. In this technique, the fre-
quency ratio, mass ratio and damping ratio of the TMD are set as the optimal parameters
to be searched, with the optimization objective being the fatigue life. When considering
the time-varying scour process, the time-varying scour depth curve is first divided into
a number of scour depths with an increment of 0.1d. For each scour depth, the fatigue
damage is calculated respectively and then the total fatigue damage in a particular du-
ration can be summarised. When the scour pit becomes deeper, the fatigue damage ac-
cumulates and finally reaches unit 1 which denotes the end of fatigue life. The simpli-
fied 4-DOF modal model incorporating scour modelling is used to generate the stress
time series. The optimization problem is formulated to determine the optimal parame-
ters of the TMD that maximize the fatigue life of the OWT support structure. The
"GlobalSearch" function in MATLAB is used to solve the optimization problem. In the
TMD optimization process, the mass ratio of the TMD is initially set to 1 %, and only
the parameters of frequency ratio and damping ratio are optimized. Subsequently, in
order to understand the optimization effect of TMD when the value of TMD mass ratio
is not fixed, a mass ratio optimization interval is provided, making the mass ratio a
variable within the optimization range.
4 **Results**
4.1 **Environmental states and load cases**
In this study, fatigue analyses are performed under 22 environmental states pro-
vided by Tempel (2006), taking into account both operational and parked conditions.
These 22 environmental states are shown in Table 4. In operating conditions, the wind
turbine withstands the aerodynamic load of the rotating rotor and the wind load of the
tower. The wind load on the rotor is calculated using the BEM theory. In parked con-
ditions, the wind turbine mainly withstands the aerodynamic load on the tower, and the
aerodynamic damping is very small. The aerodynamic loading on the blades is calcu-
lated by directly looking at the aerodynamic loading coefficient table based on the local
attack angles. The wind and wave loads are assumed to always act in the same direction
to simplify the analysis. When the mean wind speeds are above the cut-in wind speed
and below the cut-out wind speed, a  95 % wind turbine availability is assumed follow-
ing the setting in Ref (Velarde et al., 2020), meaning that the OWT does not produce
power for 5 % when the mean wind speeds are in the operating range. For a specific
combination of mean wind speed, wave period, and wave height, six different random
seed numbers are used to produce various wind fields and wave profiles, minimizing
the impact of randomness. To obtain the stress time histories at the mudline, a 700 s
simulation is conducted for each random seed, and the response in the first 100 s is
subtracted to eliminate the effect of initial transient vibration. (Capaldo and Mella, 2023;
Stieng and Muskulus, 2020).
Table 4. Environmental states, adopted from Tempel (van der Tempel, 2006).

| State | $V_w$ (m/s) | $T_z$ (s) | $H_s$ (m) | $P_{State}$ (%) | State | $V_w$ (m/s) | $T_z$ (s) | $H_s$ (m) | $P_{State}$ (%) |
|-------|------|------|------|------|-------|------|------|------|------|
| 1 | 4 | 3 | 0.5 | 3.95 | 12 | 14 | 5 | 2 | 3.26 |
| 2 | 4 | 4 | 0.5 | 3.21 | 13 | 16 | 4 | 2 | 1.79 |
| 3 | 6 | 3 | 0.5 | 11.17 | 14 | 16 | 5 | 2.5 | 3.1 |
| 4 | 6 | 4 | 0.5 | 7.22 | 15 | 18 | 5 | 2.5 | 1.74 |
| 5 | 8 | 3 | 0.5 | 11.45 | 16 | 18 | 5 | 3 | 0.8 |

| | | | | | | | | | |
|---|---|---|---|---|---|---|---|---|---|
| 6 | 8 | 4 | 1 | 8.68 | 17 | 20 | 5 | 2.5 | 0.43 |
| 7 | 10 | 3 | 0.5 | 5.31 | 18 | 20 | 5 | 3 | 1.14 |
| 8 | 10 | 4 | 1 | 11.33 | 19 | 22 | 5 | 3 | 0.4 |
| 9 | 12 | 4 | 1 | 5.86 | 20 | 22 | 6 | 4 | 0.29 |
| 10 | 12 | 4 | 1.5 | 6 | 21 | 24 | 5 | 3.5 | 0.15 |
| 11 | 14 | 4 | 1.5 | 4.48 | 22 | 24 | 6 | 4 | 0.1 |

In Table 4, $V_w$ is the wind speed, $T_z$ is the zero-crossing wave period, $H_s$ is the
wave height, and $P_{state}$ is the probability of environmental state. To investigate the ef-
fect of scour and installation of the TMD on the fatigue damage accumulation, six load
cases (LCs) are selected as shown in Table 5. LC 1 is used as the reference case, and
other cases are distinguished by different scour and TMD settings. For LC 4 to LC 6,
the initial design of the TMD with the mass ratio of 1 % is used.

Table 5. Load case definition

| LC number | TMD condition | Scour condition | LC number | TMD condition | Scour condition |
|---|---|---|---|---|---|
| LC 1 | No | No Scour | LC 4 | Enable | No Scour |
| LC 2 | No | Time-varying | LC 5 | Enable | Time-varying |
| LC 3 | No | Maximum | LC 6 | Enable | Maximum |

When considering the time-varying scour depth, for a particular time t, the time-
varying scour depth S can be predicted by the equation provided by Nakagawa et al.

(1976):

$$S = \left(\frac{t}{t_1}\right)^{0.22} D, \tag{13}$$

where D is the diameter of the monopile, $t_1$ is the reference time and can be calculated
by

$$t_1 = 29.2 \cdot \frac{D}{\sqrt{2} \cdot u} \cdot \left(\frac{\sqrt{\Delta \cdot g \cdot d_{50}}}{\sqrt{2} \cdot u - u_c}\right)^3 \cdot \left(\frac{D}{d_{50}}\right)^{1.9}. \tag{14}$$

u is the tidal velocity and taken as 0.5 m/s, $u_c$ is the critical shear velocity and taken as
0.37 m/s, g is the acceleration of gravity and taken as 9.8 m/s$^2$, $d_{50}$ is the grain size of
sea sand and taken as 0.2 mm. The parameter $\Delta = \frac{\rho_s}{\rho_w} - 1$, where $\rho_s$ is the density of
sand and taken as 2.65 g/cm$^3$, $\rho_w$ is the density of water and taken as 1 g/cm$^3$. Rudolph
et al. (Rudolph et al., 2016) provided information on the sea state and measured the
scour depth in the North Sea where the monopile N7 is situated. The measured scour
depth was fitted well for the first five years based on the time-varying scour depth pre-
diction equation shown in Eq. (13). Therefore, the data from the North Sea site can
represent a typical ocean environment with time-varying scour and is used for the cor-
relation analysis in this study.

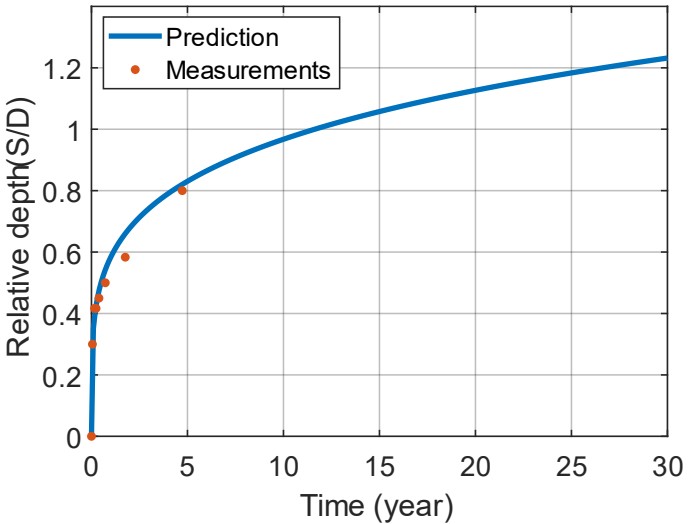


Fig. 7. Time-varying scour depth curve for pile N7 in the North Sea
When conducting an analysis with the time-varying scour, an increment of scour
depth equal to 0.1D is utilized. At a specific scour depth, the fatigue damage is calcu-
lated, and then the total fatigue damage during a longer period with varying scour
depths can be obtained by damage accumulation. According to the specification of
DNV, the maximum depth of a local scour pit formed around a pile foundation is 1.3
times the diameter of the pile. Therefore, it is assumed that the local scour pit has a
maximum scour depth of 1.3D at which the scour process achieves equilibrium.
4.2  **Scour influence on natural frequencies**
The scour of the soil around the monopile has an important effect on the natural
frequencies of the OWT. For different scour depths, the first natural frequencies ob-
tained the by the models in ABAQUS and MATLAB are compared in Fig. 8. It shows
the increase in the scour depth leads to a decrease in the first natural frequency of the
OWT. The first natural frequency is 0.265 Hz when no scour occurs, and the natural
frequency is reduced to 0.248 Hz when the depth of the scour pit reaches the maximum
depth. The first natural frequency is reduced by 6.42 % due to the maximum scour depth.
It shows that the natural frequency nearly monotonically decreases with the increase of
the scour depth. The installation of TMD also influences the natural frequency of the
OWT main structure. The TMD with a mass ratio of 1 % makes the first natural fre-
quency of the OWT main structure reduce to 0.251 Hz when no scour occurs, meaning
that the natural frequency is reduced by 5.28 %.

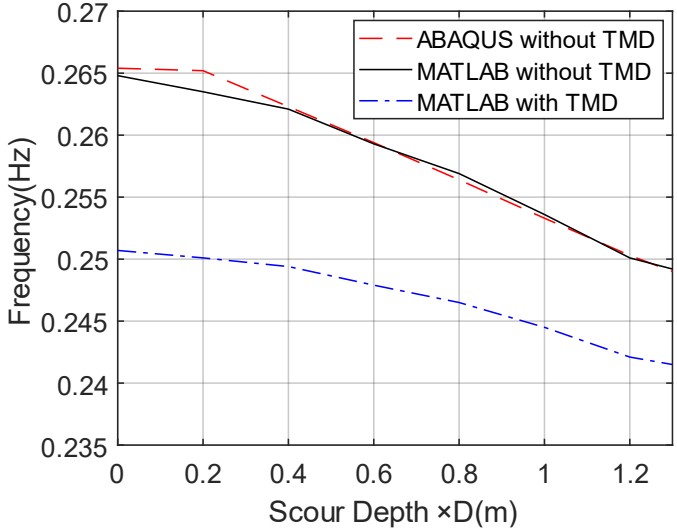


Fig. 8. Relationship between wind turbine natural frequency and scour depth

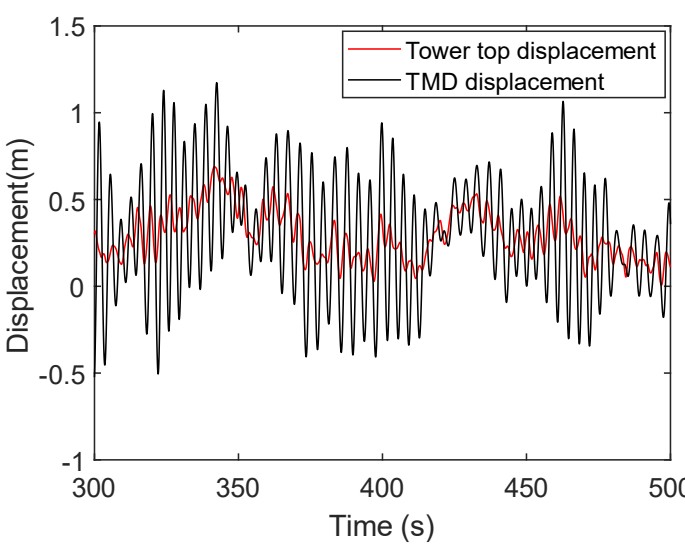


Fig. 9 Displacement of tower top and TMD under the 22nd environmental state

In the TMD design process, the feasible displacement should be considered. The

smaller the mass ratio of TMD is, the larger the feasible displacement is required. The
22nd environmental state corresponds to the greatest vibration responses of the wind
turbine tower top due to large wind speed variations and lower aerodynamic damping,
and the stroke of the TMD can be the largest. As shown in Fig. 9, the relative displace-
ment between the TMD and the tower top is much less than the inner diameter of the
wind turbine tower top in the 22nd environmental state. It shows that the stroke of the
TMD is sufficient when the mass ratio of the TMD is 1 %.
4.3 **Dynamic response analysis**

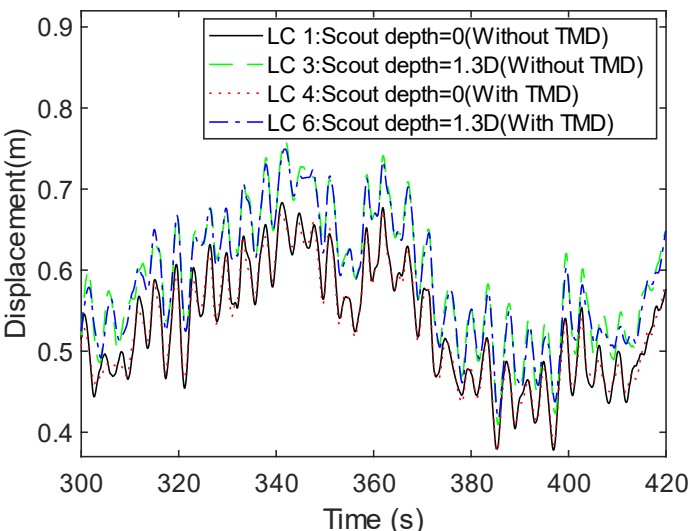


Fig. 10. Dynamic response of wind turbine under wind-wave coupled loads for four
operating conditions
When the OWT in the operating state is under the 9th environmental state which
corresponds to the rated wind speed of 12 m/s, a comparison for the tower top displace-
ments is made for LC 1, LC 3, LC 4 and LC 6, as shown in Fig. 10. These displacements
are obtained from the FE model in MATLAB described in Subsection 2.1. By compar-
ing the displacements from 300 seconds to 420 seconds for LC 1 and LC 4, it can be
found that the vibration amplitude of the tower top slightly decreases when the TMD is
installed. Moreover, by comparing the displacement responses for LC 1 and LC 3, it
can be found that the average of the displacement at the tower top increases when the
scour depth reaches 1.3D. This is because scour makes the OWT support structure be-
come more flexible. It is known that the aerodynamic damping is large when the OWT
is operating under the rated wind speed, so it is normal that the vibration mitigation
effect of the TMD is less significant in this case. The effect of the TMD is more prom-
inent for parked conditions with less aerodynamic damping. As shown in the Fig. 11,
the vibration mitigation effect of the TMD is more significant under the parked condi-
tion with 3 m/s wind speed.

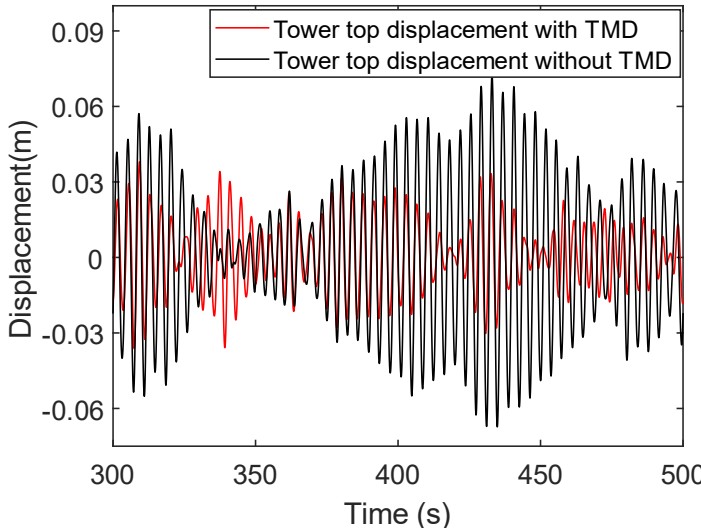


Fig. 11 The displacement response of wind turbine tower under the parked condition

with 3 m/s wind speed

4.4 **Fatigue calculation results**
In Subsection 2.4, it is mentioned that in the process of fatigue life analysis, the 4-
DOF simplified modal model is used to greatly save the calculation time. The accuracy
test of the 4-DOF modal model in generating dynamic responses is first presented in
this subsection. Under the turbulent wind field with a turbulence intensity of 11.9 %
and an average wind speed of 12 m/s, the FE model and the 4-DOF simplified modal
model are used to calculate the stress responses at the mudline for 10 minutes.

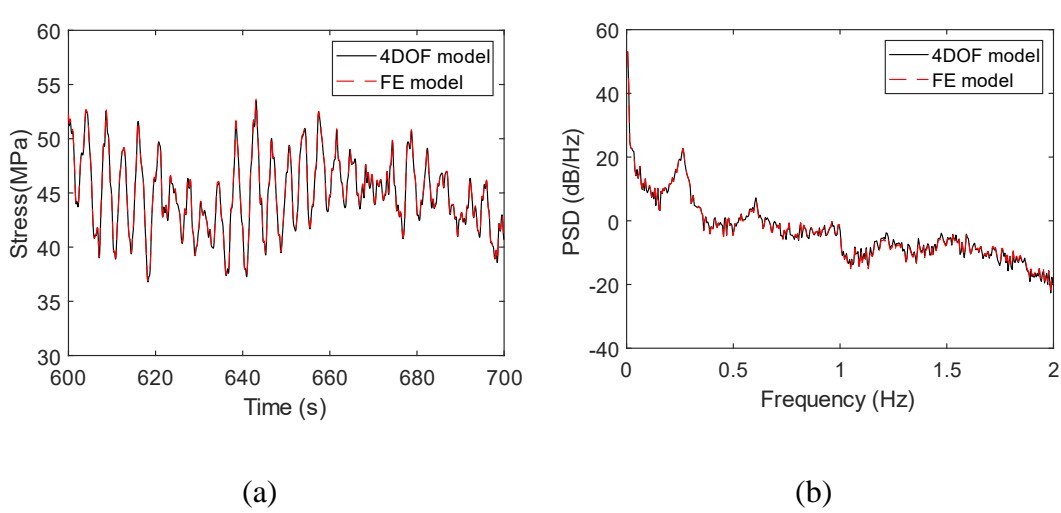

(a)                                    (b)

Fig. 12. Comparison of stresses at the mudline from the FE model and the 4-DOF

model in time domain (a) and frequency domain (b)

As shown in Fig. 12, the stress responses from these two models are very close,
confirming the accuracy of the 4-DOF modal model. The fatigue damage caused by the
FE model in 10 min is $2.108 \times 10^{-7}$, and the fatigue damage caused by the 4-DOF
model in 10 min is $2.1 \times 10^{-7}$, with an error of 0.05 %. Moreover, the calculation time
of the 4-DOF simplified modal model is only about 1/55 of that of the FE model, which
shows that the 4-DOF simplified modal model is adequate to replace the FE model
when conducting fatigue life prediction.

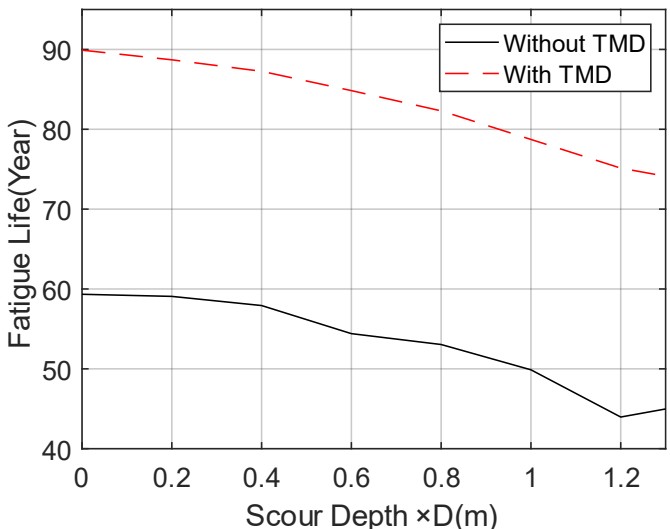


Fig. 13. Fatigue life of wind turbine with different scour depths

The 4-DOF simplified modal model is used to conduct fatigue life prediction for
the OWT support structure under LC 1 to LC 6. A 10 min simulation is conducted for
each of the six different random seed numbers to obtain stress time histories at the
mudline. The location of the hotspot selected to assess fatigue damage is where the
maximum stress occurs, specifically in the support structure cross-section at the mud-
line. Although the location in the monopile where the moment reaches its maximum
value can be below the mudline, the location at the mudline is chosen for simplicity.
Further, since the SSI is modelled in the FE model by an equivalent soil stiffness matrix,
it is not straightforward to determine the internal forces at the cross sections below the
mudline. Given the stress time series at the selected hotspot, the corresponding fatigue
damage is calculated. The fatigue damage for the combination of mean wind speed,
wave period, and wave height over 10 minutes is calculated by averaging the fatigue
damage across six random seeds. For all the 22 environment states, the 10 min fatigue
damage are calculated. The fatigue life is predicted according to the Palmgren-Miner
sum rule by combining these calculated fatigue damage and the probabilities of the
environmental states.
For various scour depths, the fatigue life of the OWT considering both operating
and parked conditions is predicted with or without TMD installation, and the results are
shown in Fig. 13. It is shown that an increase in scour depth leads to a decrease in
fatigue life, and an increasing fatigue life reduction rate can be observed when the scour
depth increases. When no scour occurs and the TMD is not installed on the OWT, the
OWT's fatigue life is 59.3 years, and the fatigue life drops to 45.0 years when consid-
ering the maximum scour depth of 1.3 D. There exist some uncertainties in the fatigue
life prediction process due to the generation of random wind field and wave profile. It
should be noted that the predicted fatigue life is much longer than the normally adopted
OWTs' design fatigue life of 25-30 years. This can be explained by the following rea-
sons. First, the maximum moment of the OWT support structure is not at the cross-
section at the mudline where the selected hotspot is located. Second, the complex wind
and wave directionality during the OWT's lifetime is simplified, which can influence
the fatigue calculation results. Third, many other operational conditions such as start-
up and shut-down phases are not considered in this study, which can also impact fatigue
damage accumulation. Moreover, the installation of the TMD greatly extends about
51.8 % of the OWT support structure's fatigue life.

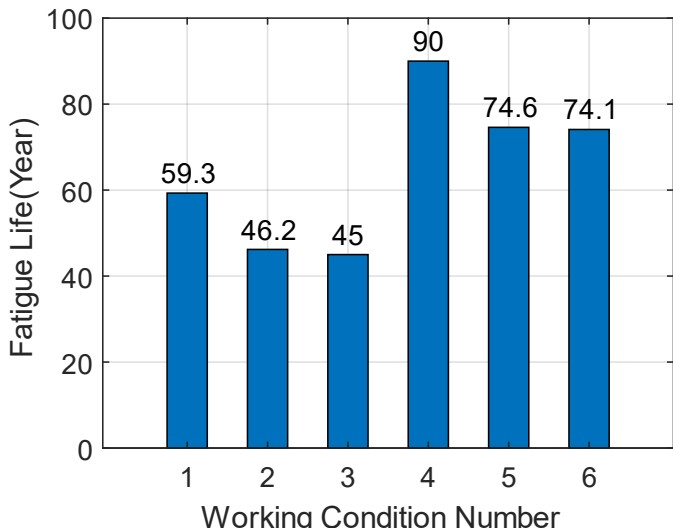


Fig. 14. Fatigue life of the wind turbine under six operating conditions
The fatigue life prediction results of the OWT are obtained for all the six LCs, as
shown in Fig. 14. The fatigue life of the reference case LC 1, which is 59.3 years, is
regarded as the reference fatigue life. It shows that the fatigue life decreases by 14.3
years, or about 24.1 %, when the scour depth is set at the maximum value of 1.3D
without applying the TMD, compared to the reference fatigue life. When considering
the time-varying scour, the fatigue life decreases by about 22.1 % from the reference
value. When comparing the results for LC 1 and LC 4, it shows the installation of the
TMD results in a significant increase in the fatigue life of the OWT, with an increase
in fatigue life of about 30.7 years, which is about 51.8 %. In LCs with the TMD installed,
the fatigue life in LC 6 decreases by about 17.7 % when the scour depth reaches 1.3D,
compared to the result in LC 4. But the fatigue life in LC 6 is still 1.25 times the refer-
ence fatigue life, which indicates that the imposition of TMD can effectively increase
the fatigue life of the OWT by reducing vibration amplitudes.
4.5  **Fatigue calculation with optimized TMD**
To compare the optimization effect and speed up the optimal parameter search
process, the mass ratio of TMD is first kept as 1 %. Before the optimization, the param-
eter ranges of the frequency ratio and damping ratio need to be defined. The optimal
frequency of the TMD is usually close to the frequency of the main structure, so the
range of the frequency ratio is chosen to be from 0.8 to 1.1 for optimization. As the
optimal damping ratio can vary in a relatively larger range, the range of the damping
ratio for optimization is chosen to be from 1 % to 30 %. The optimization of the TMD
is also conducted with the mass ratio not fixed. A range of the mass ratio from 0.001 to
0.1 is used to optimize the TMD so that the influence of the mass ratio can be evaluated.
Table 6. Optimization of TMD parameters

| Optimization method | Mass ratio range | Time-varying scour | Optimal mass ratio | Optimal frequency ratio | Optimal damping ratio | Fatigue life(Year) |
|---|---|---|---|---|---|---|
| Initial (LC 5) | 0.01 | Use | 0.01 | 0.99 | 0.061 | 74.6 |
| FOT | 0.01 | Use | 0.01 | 0.94 | 0.050 | 93.2 |
| FOT | 0.001-0.1 | Use | 0.097 | 0.92 | 0.150 | 133.2 |

The optimal parameters obtained by FOT, as well as the predicted fatigue life, are
listed in Table 6. The fatigue life for LC 5 and the parameters of the initially designed
TMD are also shown in Table 6 for comparison. It shows that when the mass ratio is
fixed at 1 %, the optimal frequency ratio is 0.94, the optimal damping ratio is 5 %, and
the final fatigue life is 93.2 years. Compared to the fatigue life with the initially

optimized TMD using the traditional method without considering scour, the fatigue life increases by 18.6 years, which is approximately 25%. It indicates that the parameter search in the optimization process is correct, and it is better to use the TMD parameter search method to design the TMD after obtaining the time-varying scour curve. When the mass ratio range is taken from 0.1 % to 10 %, the optimal mass ratio of the TMD is 9.7 %, the frequency ratio is 0.92, the damping ratio is 15 %, and the final fatigue life is 133.2 years. In this case, the fatigue life of the OWT is significantly increased primarily because of the large mass ratio. However, in practice, it may be uneconomical to implement a TMD with such a large mass ratio.

## 5  Conclusions

This study establishes a rapid numerical model that can consider the effect of scour and the installation of a TMD, and the TMD operates only in the FA direction. The model is simplified by using concentrated mass instead of RNA and ignores the non-linearity of the equivalent stiffness matrix. The established model is used to investigate the influence of scour and the installed passive structural control device on the OWT's natural frequencies and fatigue life through 22 environmental states. An optimization technique has also been developed to find the optimal parameters of the TMD considering time-varying scour. Moreover, it shows that the vibration amplitude of the OWT can be effectively reduced by the TMD. On the one hand, the results show that the TMD reduces the vibration amplitude of the tower top. On the other hand, when the scour depth reaches 1.3D, the wind turbine support structure becomes more flexible, with the displacement of the tower top increasing without TMD.

In addition, the fatigue calculation results show that the installation of the TMD significantly extends the fatigue life of the OWT, but scour can cause a reduced performance of the TMD. It is found that when scour and the scour-induced natural frequency reduction are considered during the OWT's lifetime, the performance of the initially designed TMD is not as good as the TMD optimized by the developed FOT in terms of fatigue life enhancement. Given a mass ratio of 1 %, the fatigue life can be extended by 25 % with the TMD optimized by FOT. This is because FOT can consider the effect of time-varying scour. This study focuses solely on analysing scour, but it is important to note that other factors like soil degradation can also impact the dynamic characteristics of OWTs. These factors may influence on structural control devices and the evaluation of fatigue life. Additionally, during OWTs' lifetime, the properties of installed TMDs

can also change, making the evaluation of TMDs' performance and OWTs' fatigue life
more complicated. These factors are worth investigating in the future.
**Author contribution**
Yu Cao: Investigation, Software, Validation, Formal Analysis, Writing - Original Draft
Ningyu Wu: Data Curation, Investigation
Jigang Yang: Validation, Software
Chao Chen: Conceptualization, Methodology, Writing - Review & Editing, Supervision,
Funding acquisition
Ronghua Zhu: Methodology, Resources
Xugang Hua: Writing - Review & Editing, Funding acquisition
**Competing interests**
The contact author has declared that none of the authors has any competing inter-
ests.
**Acknowledgements**
The financial supports from the National Natural Science Foundation of China (No.
52108280), Yangjiang Offshore Wind Laboratory (No. YJOWP-OF-2022A10),
the National Science Fund for Distinguished Young Scholars (No. 52025082) are
greatly appreciated.

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
