# Peer review of "Effect of scour on the fatigue life of offshore wind turbines and its prevention through passive structural control Yu Cao1, Ningyu Wu2, Jigang Yang2, Chao Chen1,3\*, Ronghua Zhu3,4\*, Xugang Hua1, 1 Key Laboratory for Bridge and Wind Engineering of Hunan Province, College of Civil Engineering, Hunan University, Changsha, China 2 Hebei Construction Investment Offshore Wind Power Co., Ltd., Tangsha"

_Wind Energy Science, 2023_

## Referee Comment (RC2)

**"Fatigue life evaluation of offshore wind turbines considering scour and passive structural control"**
**(Manuscript number: wes-2023-149)**

In this work, the influence of scour and tuned-mass dampers (TMD) on the fatigue life offshore wind turbines is investigated. Furthermore, a new approach to determine optimal parameters for the design of the TMD is proposed, which considers time-variable scour depths.

As the fatigue life is of major importance for the design of offshore wind turbines and scour and TMD both influence the fatigue life significantly, the topic is relevant for the readers of the WES journal. Moreover, most of the paper is nicely written and the argumentation is mainly clear. Nonetheless, there are several points that must be clarified or corrected.

Points to be clarified/corrected:

1) In industry, quite frequently scour protection systems are used nowadays. Hence, the topic of scour might become less relevant. It would be nice if you could briefly discuss scour protection in your introduction.
2) L. 77: I think that the statement "This study can provide a guidance for the fatigue life evaluation […]" is exaggerating, as you use a simplified fatigue life analysis and there is other work really focusing on this topic. With the second part on TMD, I totally agree, as this is the core of your work.
3) Fig. 1: In the caption of the Fig. 1, you write "scour effect" and yes, it is shown in the figure. However, it is not marked. I think it would help to mark it.
4) L. 87: What do you mean by "three-dimensional beam"? It is just a standard Euler-Bernoulli or Timoshenko beam?
5) L. 88-91: You use only a few beam elements. Is the number sufficient? Have you conducted a convergence study? Please, show it.
6) L. 83-94: Perhaps, a figure showing the FE model would help to see where the loads are applied, the TMD is positioned etc.
7) Equation 1 and 2: Are $C_T$ and $c_T$ the same (same for $K_T$ and $k_T$)?
8) L. 102: Is $u_s$ actually the displacement vector of the tower top? Isn't it the displacement vector of all nodes of the main structure?
9) L. 109-124: If I understand it correctly, you do not use a wind turbine controller when calculating the wind load from the turbulent wind field. This is a significant simplification. I do not know how important this simplification is in this context, but it might be relevant. At least, you have to discuss this simplification.
10) Section 2.3: You discuss that you use the more complex ABAQUS model and not simplified p-y curves for the derivation of your stiffness matrix. This is totally fine. However, the soil stiffness is load dependent. The load dependency is even represented by the p-y curves, but not by your stiffness matrix. You definitely have to discuss the load dependence of the soil stiffness.
11) Eq. 6: I think it should be $\mathbf{\Psi}^T M \mathbf{\Psi} \ddot{\alpha} + \mathbf{\Psi}^T C \mathbf{\Psi} \dot{\alpha} + \mathbf{\Psi}^T K \mathbf{\Psi} \alpha = \mathbf{\Psi}^T F$ and not $\mathbf{\Psi}^T M \mathbf{\Psi} \ddot{u} + \mathbf{\Psi}^T C \mathbf{\Psi} \dot{u} + \mathbf{\Psi}^T K \mathbf{\Psi} u = \mathbf{\Psi}^T F$ as the transformation is $\mathbf{\Psi}\alpha = u.$
12) Eq. 8 is not sufficiently clear. For example, x and y are not explained. Furthermore, the element shape functions are neither given nor explained. It is not stated that the shape functions refer to the original FE model, i.e., the model before applying the modal reduction.
13) L. 243: "t is the thickness at which cracks may grow"; are you sure that this statement is correct? Isn't t the actual thickness of the pile?
14) Table 3: A SCF is given. However, you never state how you use it.
15) L. 251: $N_c$ is obtained by the rainflow counting? I think $N_c$ has to be defined before the rainflow counting can start, as it is the number of bins, the rainflow counting sorts the cycles into. Which value do you use for $N_c$?

16) L. 272-274: I would be careful when stating that the fore-aft mode is the most important one. For large monopile and significant wind-wave-misalignments, side-to-side modes can be more critical with respect to fatigue, as the aerodynamic damping is lower in side-to-side direction.

17) L. 275-276: If the TMD is in the tower, is it still rotating, when the RNA is yawed? Otherwise, I wonder how the TMD can always be aligned with the fore-aft direction.

18) L. 290: Why did you choose 1% for the mass ratio and not any other value?

19) Section 3.2: Perhaps, it would help to give a short example demonstrating how $c_T$ and $k_T$ change if the eigenfrequency drops to, for example, 0.26 Hz due to scour.

20) Figure 6: Is the equivalent stiffness matrix actually "added" to the 4DOF model? I thought that it is added to the MATLAB FE model.

21) Figure 6: The parts on "Divided by 0.1D" and "Scour depth plus 0.1D" are completely unclear at this stage. They become a bit clearer later on, but I think some explanation or at least a reference to a later section is needed here. Otherwise, the reader is lost.

22) L. 309: Here, you state that the mass ratio is a variable. Before, you just select a value, i.e., 1%, for it. Later, you do both. This is quite confusing when you read the paper for the first time. Perhaps, it would help to elaborate a bit more on it it (see comment 17 as well).

23) L. 313: You state that you use Fmincon. First of all, Fmincon is just a MATLAB routine. What is actually interesting is which optimization algorithm is used. If I remember correctly, Fmincon uses a local optimization algorithm. Is this sufficient? Have you looked at the objective space and it is rather smooth without local minima? Otherwise, a global optimization algorithm might be more appropriate.

24) L. 318: You state that you model operational and parked conditions. What is the difference between these two in your simplified model without a controller? Are only the wind loads different or do you also change the inertia of the RNA etc.? In reality, even the first fore-aft bending eigenfrequencies of the entire turbine are slightly different in operational and parked conditions.

25) Table 4: $V_w$, $T_z$, $H_s$ and $P_{state}$ are not explained.

26) L. 317-319: You state that you have 22 environmental states for operational and parked situations. However, only in line 401, you start to explain that you run six 10-min simulations for each condition. This should already be stated here. Furthermore, two questions are unanswered in my opinion: 1) What is the total number of simulations? Is it $2 \times 22 \times 6$ (operating/parked x environmental states x seeds)? 2) Do you remove some time at the beginning of each simulation to remove initial transients? If yes, how much?

27) L. 338-339: What are the values for $d_{50}$, $\rho_s$ and $\rho_w$ you use?

28) Fig. 6: Perhaps, it would be nice if you add two other graphs to this figure for the TMD (ABAQUS + MATLAB) or at least one for MATLAB if you do not have the TMD implemented in ABAQUS.

29) L. 371: Is this case an operating or a parked case?

30) L. 379: You state that the effect is more prominent for other operating conditions. First, I think that is it especially more prominent for parked conditions with less aerodynamic damping. And second, please show a case, where the effect is more prominent. You can just add a second figure.

31) Fig. 7: I can hardly read this figure in greyscale. Please, enlarge it and think about clearly different line styles (and perhaps also thicker lines).

32) Fig. 7: In the caption, you write "four operating conditions". I think it should be "four load cases".

33) Section 4.4: You visually compare time series and spectra for the 4DOF and the FE model. This is a good starting point. However, frequently, you cannot see the differences leading to different fatigue lifetimes in these plots immediately. Hence, it would be good if you could also calculate the damage value $D_k$ (Eq. 10) for this time series and the two models. This would be an objective comparison.

34) L. 434 and 445: How does these two values (50% and 62%) fit together?

35) L. 461: You use your "standard" value of 1% as a boundary value for the optimization. This is not a good approach, as it excludes all values below the "standard" value. Please, either repeat your optimization with another boundary value or justify your choice.

36) Table 6: I think that it would help if you name the first row "Initial (LC 5)". This would make things much clearer.

37) L. 471: You state that your results indicate that "considering time-varying scour depth" is beneficial. However, you cannot know this from your results, as you directly compare "fixed" TMD parameters with optimized ones which consider time-varying scour depth. What you do not compare are optimized TMD parameters for the maximum scour depth. I can imagine that these are quite similar to the ones you have determined for the time-varying scour depth. Hence, perhaps, the benefit is just due to the optimization. Therefore, you should either include TMD parameters that are optimized for the maximum scour depth in your analysis, or you should it least discuss this aspect here.

38) Conclusions: You should clearly state your major simplifications, e.g., simplified lifetime calculation with just 22 environmental states, no controller, TMD only in fore-aft direction etc.

Typos etc.:

1) As you can see in the following, there are some typos and inconsistencies. As I have definitely not found all of them, I recommend a thorough proof reading.

2) Please, check your citation style, e.g., in line 43, it should be "Sørensen and Ibsen, 2013". The other names are given names.

3) L. 49 and others: "damage" and not "damages". There is no plural of "damage" in the context of structural engineering.

4) L. 83: "An FE model" and not "A FE model"

5) L. 83: "a monopile-supported OWT" and not "an monopile-supported OWT"

6) Table 1: "Rated wind speed" not "Rated wind Speed"

7) L. 129: "in Shirzadeh et al. (2013)" and not "in Ref. (Shirzadeh et al., 2013)"

8) Table 2: "kN/m³" and not "kN/m3"

9) L. 227: Remove "the" before "Eq. (6)

10) L. 287 "and $\xi_{opt}$ is the" not "and is the"

11) L. 296: "885 $Ns/m$" and not "885 $N \cdot s/m$"

12) L. 333: Remove "was used"

13) L. 406: I think it should be "where the maximum stress is reached".

14) L. 484: Where is the "on the one hand"? You just use "on the other hand.

15) L. 578: "Patil" not "patil".

16) L. 594: See typo comment 2.

17) L. 628 "van der Tempel, J. (2006)" and not „Tempel, J. van der. (2006)"

---

## Author Comment (AC1)

**(Manuscript number: wes-2023-149)**

**Fatigue life evaluation of offshore wind turbines considering scour and passive structural control**

**By Yu Cao, Ningyu Wu, Jigang Yang, Chao Chen, Ronghua Zhu, Xugang Hua**

**Overall Response:**

We are very grateful to the Editor and Reviewers for their constructive comments on this manuscript. We have revised and improved the manuscript based on the comments and clarified the issues brought up in the paper. In the following sections, point-by-point responses to the comments were provided. The original comments are in italics. The authors' responses are highlighted in blue. The corresponding changes are highlighted in red in the revised manuscript.

**Other comments:**

**Reviewer #1**

*Comment 1: In Section 2.1, the combined soil and structural damping is presented as 1% without elaboration. This figure seems quite low for soil. Also, how is this implemented in Rayleigh damping?*

**Response:** In this study, both soil damping and structural damping are combined together and set as Rayleigh damping. The Rayleigh mass coefficients and Rayleigh stiffness coefficients are calculated by the follow equation.

$$\alpha_1 = \alpha_2 = \frac{\zeta_c}{\frac{1}{2\omega} + \frac{\omega}{2}}$$

where $\alpha_1$ and $\alpha_2$ are the Rayleigh mass and stiffness coefficients, respectively $\omega$ is the natural frequency of the 1st fore-aft mode, and $\zeta_c$ is the combined damping ratio.

According to the authors' previous studies on damping in monopile-supported OWTs (Chen and Duffour, 2018) , the structural damping is in the range of 0.2%-1.5%, and the soil damping is between 0.17% -1.3%, so a total damping ratio of 1% for the first mode is a little small but still a rational quantify to define the damping.

**Revised manuscript:**

**L110-L115:** The damping matrix is applied by means of Rayleigh damping, and the combined damping ratio of soil damping and structural damping is assumed to be 1% (Chen and Duffour, 2018) . The Rayleigh mass and stiffness coefficients $\alpha_1$ and $\alpha_2$ are defined by $\alpha_1 = \alpha_2 = \frac{\zeta_C}{\frac{1}{2\omega}+\frac{\omega}{2}}$. $\omega$ is the natural frequency of the first fore-aft mode, and $\zeta_C$ is the combined damping ratio. The RNA is represented by a lumped mass at the tower top.

***Comment 2***: *In Equations 1 and 2, there is ambiguity regarding dimensions. How are "C_T" and "K_T" defined, and how do they differ from "c_T" and "k_T"? Since the TMD should have one DOF, how is the "u_T" vector defined?*

**Response:** In Equations 1 and 2, there is indeed an ambiguity about the matrix dimension problem. Equations 1 and 2 have been modified to clarify the matrix notations as follows:

$$\mathbf{M_s}\ddot{\mathbf{U}}_s + \mathbf{C_s}\dot{\mathbf{U}}_s + \mathbf{K_s}\mathbf{U}_s + \mathbf{C_T}(\dot{\mathbf{U}}_s - \dot{\mathbf{U}}_T) + \mathbf{K_T}(\mathbf{U}_s - \mathbf{U}_T) = \mathbf{F}_{\text{wind}} + \mathbf{F}_{\text{wave}}, \quad (1)$$

$$m_T\ddot{u}_T + c_T(\dot{u}_T - \dot{u}_{s-top}) + k_T(u_T - u_{s-top}) = 0, \quad (2)$$

In Equation 1, the equation is constructed based on all the nodes of the wind turbine structure plus the tunned mass damper. $\mathbf{M_s}, \mathbf{C_s}, \mathbf{K_s}, \mathbf{C_T}, \mathbf{K_T}$ have the same dimensions, and $\mathbf{U_s}, \mathbf{U_T}$ have the same dimensions. In Equation 2, the equation is constructed for the TMD node at the top of the tower, $m_T, c_T, k_T$ are the mass, damping

and stiffness of the TMD, and $u_T, u_{s-top}$ are the displacement of the TMD and the displacement of the top node respectively.

The definitions of $\mathbf{C}_T, \mathbf{K}_T, c_T, k_T$ are explained as follows: $\mathbf{C}_T$ is a matrix containing $c_T$, $\mathbf{K}_T$ is a matrix containing $k_T$, and their relations are as follows:

$$\mathbf{C}_T = \begin{bmatrix} 0 & \cdots & 0 \\ \vdots & \ddots & \vdots \\ 0 & \cdots & c_T \end{bmatrix}, \mathbf{K}_T = \begin{bmatrix} 0 & \cdots & 0 \\ \vdots & \ddots & \vdots \\ 0 & \cdots & k_T \end{bmatrix}, \mathbf{U}_T = \begin{bmatrix} 0 \\ \vdots \\ u_T \end{bmatrix}, \mathbf{U}_s = \begin{bmatrix} u_{s-1} \\ \vdots \\ u_{s-top} \end{bmatrix}$$

$u_T$ is the absolute displacement of TMD at the top of the tower. In this paper, TMD moves in the FA direction and does not move in the SS direction, so $u_T$ is the displacement vector of TMD in the FA direction with respect to time.

**Revised manuscript:**

**L118-L130:** The TMD is mounted on the top of the tower, and the effect of the TMD is considered by adding its mass, damping, and stiffness terms at relevant positions in the local mass, damping, and stiffness matrices of the beam element representing the tower top. The equation of motion of the OWT main structure is:

$$\mathbf{M}_s\ddot{\mathbf{U}}_s + \mathbf{C}_s\dot{\mathbf{U}}_s + \mathbf{K}_s\mathbf{U}_s + \mathbf{C}_T(\dot{\mathbf{U}}_s - \dot{\mathbf{U}}_T) + \mathbf{K}_T(\mathbf{U}_s - \mathbf{U}_T) = \mathbf{F}_{\text{wind}} + \mathbf{F}_{\text{wave}}, \quad (1)$$

where $\mathbf{M}_s, \mathbf{C}_s, \mathbf{K}_s$ are the mass, damping and stiffness matrices of the main structure. $\mathbf{C}_T, \mathbf{K}_T$ are matrices with same dimensions containing $c_T, k_T$. $\mathbf{U}_s$ is the displacement vector of the main structure, and $\mathbf{U}_T$ is the displacement vector containing $u_T$. $\mathbf{F}_{\text{wind}}$, $\mathbf{F}_{\text{wave}}$ are the aerodynamic and wave load vectors. The equation of motion for the TMD can be represented by

$$m_T\ddot{u}_T + c_T(\dot{u}_T - \dot{u}_{s-top}) + k_T(u_T - u_{s-top}) = 0, \quad (2)$$

where $m_T, c_T, k_T$ are the mass, damping and stiffness of the TMD, $u_T, u_{s-top}$ are the displacement of the TMD and the displacement of the top node. The modelling of SSI is realized by an equivalent stiffness matrix, which will be introduced in detail subsequently in Section 2.3.

**Comment 3**:*In Section 2.4, the forces on the right side of Equation 6 are nonlinear, and their truncation essentially linearizes them. Please elaborate on this and clarify this step in the manuscript.*

**Response:** Sorry for the confusion. In Equation 6, before the modal reduction to a 4-DOF model, it indeed requires linearizing the aerodynamic forces from the rotor on the tower top and the hydrodynamic forces, which are not mentioned in the manuscript. We have added explanations for the force linearization.

**Revised manuscript:**

**L244-L249:** The total aerodynamic forces from the rotor applied on the tower top node are linearized to the sum of a term corresponding to the forces for an assumed rigid tower, plus a term proportional to the tower top linear and angular velocities. The hydrodynamic forces are linearized by ignoring the relatively small monopile vibrations. The details for force linearization can be found in the authors' previous studies (Chen, Duffour, Fromme, et al., 2021).

**Reference:**

Chen, C., Duffour, P., Fromme, P. and Hua, X. (2021). Numerically efficient fatigue life prediction of offshore wind turbines using aerodynamic decoupling. Renew. Energy, 178, 1421–1434. https://doi.org/10.1016/j.renene.2021.06.115

**Comment 4**:*According to Section 3, the studied TMD operates in the FA direction, and the SS direction is uncontrolled. This should be mentioned in the Abstract, Introduction, and Conclusion sections.*

**Response:** The authors have revised the abstract, introduction and conclusion, adding the explanation that TMD operates in the FA direction, and the SS direction is uncontrolled.

**Revised manuscript:**

**L24-L28: Abstract:** This optimization technique aims at finding optimal parameters of the TMD which maximizes the fatigue life of a hotspot at the mudline, and effect of time-varying scour can be considered. This study assumes the TMD operates in the FA direction, and the vibration in the SS direction is uncontrolled.

**L85-L86: Introduction:** The TMD operates in the FA direction and does not work in the SS direction.

**L551-L552: Conclusions:** This study establishes a rapid numerical model which can consider the effect of scour and installation of a TMD, and the TMD operates only in the FA direction.

*Comment 5:* *The mass ratio of the TMD, chosen as 1% in the study, is quite low and could result in unfeasible TMD displacement. This effect should be discussed.*

**Response:** Thank you for your suggestion. The authors find that there is indeed a lack of discussion about whether the feasible displacement of TMD is sufficient. According to many engineering practices, TMDs with a mass ratio of 1%-2% can effectively suppress the wind-induced vibration of high-rise building structures. Moreover, previous studies have found that the TLCD with a mass ratio of 1% and the TMD with a mass ratio of 2% can effectively suppress vibration (Colwell and Basu, 2009; Lackner and Rotea, 2011b; R. Zhang et al., 2019). In the study, the authors considered that excessive mass would lead to increased construction cost and difficulty, as well as the excessive change of the inherent characteristics of the original structure, so the mass ratio of TMD was chosen to be 1%. And according to your comments, the authors studied feasible displacements with a mass ratio of 1%. The authors have calculated the displacements of the top of the wind turbine tower and TMD under the wind speed of 22m/s. It is found that the relative displacement between TMD and tower top is smaller than the inner diameter of tower top, indicating that there is no unfeasible displacement.

**Revised manuscript:**

**L325-L330:** Considering that excessive mass will lead to increased construction costs and difficulties and changes in the inherent characteristics of the original structure, the mass ratio of the TMD system to the main structure is first selected to be 1%. Moreover, previous studies have found that TLCD with a mass ratio of 1% and TMD with a mass ratio of 2% can effectively suppress vibration (Colwell and Basu, 2009; Lackner and Rotea, 2011b; R. Zhang et al., 2019).

**L426-L433:** In the TMD design process, the feasible displacement should be considered. The smaller the mass ratio of TMD is, the larger the feasible displacement is required. The 22nd environmental state corresponds to the greatest vibration responses of the wind turbine tower top due to large wind speed variations and lower aerodynamic damping, and the stroke of the TMD could be the largest. As shown in the Fig. , the relative displacement between the TMD and the tower top is much less than the inner diameter of the wind turbine tower top in the 22nd environmental state. It shows that the stroke of the TMD is sufficient when the mass ratio of TMD is 1%.

[Figure]

Fig. 9 Displacement of tower top and TMD under the 22nd environmental state

**Reference**

Colwell, S. and Basu, B. (2009). Tuned liquid column dampers in offshore wind turbines for structural control. Engineering Structures, 31(2), 358–368. https://doi.org/10.1016/j.engstruct.2008.09.001

Lackner, M. A. and Rotea, M. A. (2011b). Structural control of floating wind turbines.

Mechatronics, 21(4), 704–719. https://doi.org/10.1016/j.mechatronics.2010.11.007

Zhang, R., Zhao, Z. and Dai, K. (2019). Seismic response mitigation of a wind turbine tower using a tuned parallel inerter mass system. Engineering Structures, 180, 29–39. https://doi.org/10.1016/j.engstruct.2018.11.020

**Comment 6:** *Section 4.5 presents additional results with the TMD (FOT), optimized considering overall fatigue performance. These results should be integrated into Section 4.4 and presented alongside the TMD optimized for the initial state of the structure. This would allow, for example, the inclusion of a third and fourth curve in Figure 9, showing results incorporating TMDs optimized by FOT."*

**Response:** The author divided the contents of 4.4 and 4.5 into two sections, mainly due to the different main contents of the two sections. In section 4.4, the influence of scour depth changes on the fatigue life of wind turbines with or without coupling TMD is mainly studied. In section 4.5, the TMD parameter optimization based on the overall fatigue life of the wind turbine is mainly studied when the wind turbine is at a given mass ratio or a given mass ratio interval considering time-varying scour depth.

It is not reasonable to add the optimization results of TMD to Figure 9. In section 4.5, the TMD parameter optimization based on the overall fatigue life of the wind turbine is performed according to the time-varying scour depth. The fatigue life calculation for the parameters of the initial TMD of the structure is shown in Table 6. In this section, the scour depth is constantly changing, and the relation curve between scour depth and time refers to the scour curve of N7 single pile in the North Sea in Figure 5. The results show a fatigue life at different scour depths, so they cannot be added to Figure 9.

**Comment 7:** *Title: Consider the alternative, "Effect of Scour on the Fatigue Life of Offshore Wind Turbines and Its Prevention Through Passive Structural Control."*

**Response:** Thank you for your suggestion. The alternative topic you provided not only shows the influence of scour on fatigue life of wind turbines, but also further highlights the prevention of fatigue damage caused by scour of wind turbines through passive control. We think your proposed alternative title really better fits to the research content of this article. So, we decide to change the article title according to your suggestion.

**Revised manuscript:**

**L1-L2: Title: Effect of scour on the fatigue life of offshore wind turbines and its prevention through passive structural control**

**Comment 8:** *Abstract: The text requires editing. Some sentences are difficult to understand and contain grammatical errors. For example, ""... either by a traditional method or a newly developed ..."" should be ""... both a traditional method and a newly developed ... are presented.". Additionally, the sentence "... scour can decrease the fatigue life by about 26%, and the TMD can ... increase the fatigue life," lacks complete information about the effect of TMD. Also, the tenses change inconsistently throughout the text.*

**Response:** There are indeed grammatical errors and difficulties in understanding the sentences in the abstract. According to your valuable comments, the authors have modified the grammar and sentence to make the expression of the content more understandable and convenient for readers to read.

**Revised manuscript:**

**L17-L34: Abstract:** Offshore wind turbine (OWT) support structures are exposed to the risk of fatigue damages and scour, and this risk can be effectively mitigated by installing structural control devices such as tuned mass dampers (TMDs). However, time-varying scour altering OWTs' dynamic characteristics has an impact on the TMD design and fatigue life, which was rarely studied before. In this paper, a simplified modal model is used to investigate the influence of scour and a TMD on the fatigue life

evaluation of a 5 MW OWT's support structure, and a traditional method and a newly developed optimization technique are both presented to obtain TMD parameters. This optimization technique aims at finding optimal parameters of the TMD which maximizes the fatigue life of a hotspot at the mudline, and effect of time-varying scour can be considered. This study assumes the TMD operates in the FA direction, and the vibration in the SS direction is uncontrolled. Results show that scour can decrease the fatigue life by about 24.1%, and the TMD can effectively suppress vibration and increase the fatigue life. When the scour depth reaches 1.3 times the pile diameter, the TMD with a mass ratio of 1% can increase the fatigue life of OWT's support structure by about 64.6%. Further, it is found that the fatigue life can be extended by 25% with the TMD optimized by the proposed optimization technique, compared to that with the traditionally optimized TMD which does not take the change of dynamic characteristics into account.

**Comment 9:** *In the Introduction, the literature review includes vibration control systems only up to 2021. Recent years have seen accelerated developments, including floating and monopile OWTs. Presenting these up-to-date examples would more clearly define the research gap.*

**Response:** The author fully agrees with you about the lack of the latest research status in recent years in the introduction and has revised and improved the introduction according to your suggestions.

**Revised manuscript:**

**L38-L70: Introduction:** With the continuous development of large-size fixed-bottom OWTs, local scour and scour protection of pile foundation have become a common issue (L. Wang et al., 2020; X. Wang et al., 2019; F. Zhang et al., 2022). Scour have a significant impact on dynamic characteristics, vibration magnitudes, and thus fatigue life of OWTs under wind and wave loads. On the one hand, the action of currents and waves causes local scour pits around pile foundations, which reduces the burial depth

of pile foundations. This phenomenon usually causes a reduction in natural frequencies of OWTs and changes in other dynamic characteristics, possibly leading to resonance, large amplitude stress cycles and fatigue damage when one of natural frequencies is close to the rotational frequency of the blades (Sørensen and Ibsen, 2013). On the other hand, current scour protection measures cannot completely avoid scour and have their own shortcomings. For example, armouring protection has the disadvantages that the projectile cannot be accurately cast in complex sea conditions and is easy to be washed away (G. Wang et al., 2023; F. Zhang et al., 2023). Flow-altering protection has the disadvantages of high cost and changing the dynamic characteristics of the foundation (Tang et al., 2023). As offshore structures, wind turbines are vulnerable to corrosion from seawater, which makes the fatigue problem worse (Amirafshari et al., 2021). Thus, the scour-induced changes in dynamic characteristics and risk in resonance inevitably induce a further increase in fatigue damage and deserve in-depth research (Mayall et al., 2018).

Many researchers have studied the effect of scour on fatigue damage accumulation in OWTs. For instance, Tempel et al. (2006) investigated the frequency and fatigue of piles under different scour depths and concluded that scour has a little effect on the natural frequencies but a great effect on fatigue damage. Zhang et al. (2021) found that scour depth has a significant influence on monopile impedance. Rezaei et al. (2018) showed that scour leads to an increase in the maximum bending moment of the monopile and a shortening of the fatigue life. To mitigate the fatigue damage in OWTs, installing structural control devices is an effective way. It was demonstrated that TMDs have a positive effect on reducing vibration amplitudes of wind turbine systems (Lackner and Rotea, 2011a; Dinh and Basu, 2015; Lu et al., 2023; Aydin et al., 2023). Dai et al. (2021) conducted a shaker experiment using a scaled wind turbine model and showed that the installed TMD can suppress the vibration of the structure more effectively considering soil-structure interaction (SSI).

***Comment 10***:*Again in the Introduction, the paper's contribution should be stated in the present tense: "In this study, ABAQUS is used ..." Furthermore, it is unclear what the FE model in MATLAB includes. The text mentions considering the scour effect, but the previous sentence stated ABAQUS was used for SSI.*

**Response:** According to your comments, the authors have modified the tenses. There is ambiguity about the description of the MATLAB model and the presentation of this part is modified in the original manuscript. In the wind load module, turbulent wind and constant wind can be generated according to demand, and in the hydrodynamic module, regular and irregular wave loads can be generated. The TMD module can set different parameters, and the beam element method is used to build wind turbine models of different sizes in the structure module. Regarding the scour effect mentioned in MATLAB, it is achieved by means of an equivalent stiffness matrix derived from the ABAQUS model.

**Revised manuscript:**

**L77-L86:** The purpose of this study is to explore the effect of scour on the fatigue life of wind turbine structures and the control effect of TMD on the fatigue life of wind turbine structures under scour conditions. The authors use a 5 MW single-pile wind turbine as a case study to carry out related research. In this study, ABAQUS is used to establish a detailed SSI model with different scour depths. A finite element model considering wind loads and TMD was established in MATLAB, and the scour effect is considered by establishing a relationship with the ABAQUS model by means of the equivalent stiffness matrix. And the finite element model is simplified to a modal model for fast prediction of fatigue life. The TMD operates in the FA direction and does not work in the SS direction.

***Comment 11***:*The paragraph mentioned above should first introduce the objectives of the study. Currently, it summarizes only the methodology. The overall purpose should be clarified as: "The aim of the present study is to ... (investigate the effect of scour ...*

*introduce an optimization method for TMDs ... through a case study involving a monopile 5 MW wind turbine.)"*

**Response:** The author agrees with you and has adjusted the content of the introduction. The author first introduces the purpose of the study, and then introduces the method of the study.

**Revised manuscript:**

**L77-L92:** The purpose of this study is to explore the effect of scour on the fatigue life of wind turbine structures and the control effect of TMD on the fatigue life of wind turbine structures under scour conditions. The authors use a 5 MW single-pile wind turbine as a case study to carry out related research. In this study, ABAQUS is used to establish a detailed SSI model with different scour depths. A finite element model considering wind load and TMD was established in MATLAB, and the scour effect is considered by establishing a relationship with the ABAQUS model by means of the equivalent stiffness matrix. And the finite element model is simplified to a modal model for fast prediction of fatigue life. The TMD operates in the FA direction and does not work in the SS direction. This study investigates the effect of different scour depths on the performance of the TMD and the fatigue life of a 5 MW OWT's support structure including a tower and a monopile foundation, and the optimization of the TMD's parameters considering time-varying scour depths to maximum fatigue life is also presented. This study provides some knowledge of the effects of the time varying scour and the TMD on the fatigue life of wind turbines, as well as a new TMD design method targeting at enhancing fatigue resistance.

**Comment 12:** *At the end of the Introduction, the structure of the paper should be outlined: "The remainder of the paper is organized as follows: Section 2 describes ..."*

**Response:** The author highly appreciates your comments. According to your comments, the author added an overview of the structure of the paper at the end of the introduction,

providing a systematic overview of the content of each chapter.

**Revised manuscript:**

**L92-L96:** The rest of the paper is organized as follows: Section 2 introduces the numerical models used in the research. Section 3 introduces the traditional TMD design method and the newly developed parameter optimization method. Section 4 describes the load cases for the fatigue analysis, the analysis results of this study and the TMD parameter optimization results. Section 5 concludes the study.

***Comment 13:*** *In Figures 9 and 10, adding a grid or labeling significant values on both curves would be helpful.*

**Response:** Thank you for your suggestion. The author highly appreciates your comments. According to your comments, the author modified Figure 9 and Figure 10 by adding grid lines to the graph to make the results better presented and easier for readers to read.

**Revised manuscript:**

[Figure]

Fig. 13. Fatigue life of wind turbine with different scour depths

[Figure]

Fig. 14. Fatigue life of the wind turbine under six operating conditions

**Comment 14:** *In Figures, try to use for better resolution.*

**Response:** Thank you for your suggestion. The author highly appreciates your comments. According to your comments, the authors have altered all the pictures.

**Revised manuscript:**

[Figure]

Fig. 7. Time-varying scour depth curve for pile N7 in the North Sea

[Figure]

Fig. 8. Relationship between wind turbine natural frequency and scour depth

[Figure]

Fig. 10. Dynamic response of wind turbine under wind-wave coupled loads for four operating conditions

[Figure]

(a)                                                    (b)

Fig. 12. Comparison of stresses at the mudline from the FE model and the 4-DOF model in time domain (a) and frequency domain (b)

***Comment 15:*** *There are quite a few self-citations (e.g., Chen 2018- Chen 2021: 6 times). Please, check the necessity of these references.*

**Response:** The author highly appreciates your comments. According to your comments, the authors have further checked the self-citations, and found that the fourth and the fifth of the six self-citations are repeated citations, and the repeated citation has been removed. The self-citations are explained as follows:

The paper "Modelling wind turbine tower-rotor interaction through an aerodynamic damping matrix" is cited mainly because the wind load calculation method and aerodynamic linearization technology adopted in this study follows the method in this reference.

The authors adopt the values of damping ratio is based on previous research achievements, so [Modelling damping sources in monopile-supported offshore wind turbines] is necessary. The details for force linearization can be found in [Identification of aerodynamic damping matrix for operating wind turbines]. The calculation method of wind load and the establishment method of simplified modal model refer to the previous paper: [Numerically efficient fatigue life prediction of offshore wind turbines using aerodynamic decoupling].

**Revised manuscript:**

**References**

Chen, C. and Duffour, P. (2018). Modelling damping sources in monopile-supported offshore wind turbines. Wind Energy, 21(11), 1121–1140. https://doi.org/10.1002/we.2218

Chen, C., Duffour, P., Dai, K., Wang, Y. and Fromme, P. (2021). Identification of aerodynamic damping matrix for operating wind turbines. Mech. Syst. Signal Process., 154, 107568. https://doi.org/10.1016/j.ymssp.2020.107568

Chen, C., Duffour, P. and Fromme, P. (2020). Modelling wind turbine tower-rotor

interaction through an aerodynamic damping matrix. J. Sound Vib., 489, 115667. https://doi.org/10.1016/j.jsv.2020.115667

Chen, C., Duffour, P., Fromme, P. and Hua, X. (2021). Numerically efficient fatigue life prediction of offshore wind turbines using aerodynamic decoupling. Renew. Energy, 178, 1421–1434. https://doi.org/10.1016/j.renene.2021.06.115

---

## Author Comment (AC2)

**Reply to Reviewers' and Editor's Comments**

**(Manuscript number: wes-2023-149)**

**Fatigue life evaluation of offshore wind turbines considering scour and passive structural control**

**By Yu Cao, Ningyu Wu, Jigang Yang, Chao Chen, Ronghua Zhu, Xugang Hua**

**Overall Response:**

We are very grateful to the Editor and Reviewers for their constructive comments on this manuscript. We have revised and improved the manuscript based on the comments and clarified the issues brought up in the paper. In the following sections, point-by-point responses to the comments were provided. The original comments are in italics. The authors' responses are highlighted in blue. The corresponding changes are highlighted in red in the revised manuscript.

**Reviewer #2**

*Comment 1:In industry, quite frequently scour protection systems are used nowadays. Hence, the topic of scour might become less relevant. It would be nice if you could briefly discuss scour protection in your introduction.*

**Response:** Thank you for your suggestion. According to your comments, the authors have revised the original manuscript and added the research progress of scour protection.

**Revised manuscript:**

**L38-L70:** With the continuous development of large-size fixed-bottom OWTs, local scour and scour protection of pile foundation have become a common issue (L. Wang et al., 2020; X. Wang et al., 2019; F. Zhang et al., 2022). Scour have a significant impact on dynamic characteristics, vibration magnitudes, and thus fatigue life of OWTs

under wind and wave loads. On the one hand, the action of currents and waves causes local scour pits around pile foundations, which reduces the burial depth of pile foundations. This phenomenon usually causes a reduction in natural frequencies of OWTs and changes in other dynamic characteristics, possibly leading to resonance, large amplitude stress cycles and fatigue damage when one of natural frequencies is close to the rotational frequency of the blades (Sørensen and Ibsen, 2013). On the other hand, current scour protection measures cannot completely avoid scour and have their own shortcomings. For example, armouring protection has the disadvantages that the projectile cannot be accurately cast in complex sea conditions and is easy to be washed away (G. Wang et al., 2023; F. Zhang et al., 2023). Flow-altering protection has the disadvantages of high cost and changing the dynamic characteristics of the foundation (Tang et al., 2023). As offshore structures, wind turbines are vulnerable to corrosion from seawater, which makes the fatigue problem worse (Amirafshari et al., 2021). Thus, the scour-induced changes in dynamic characteristics and risk in resonance inevitably induce a further increase in fatigue damage and deserve in-depth research (Mayall et al., 2018).

Many researchers have studied the effect of scour on fatigue damage accumulation in OWTs. For instance, Tempel et al. (2006) investigated the frequency and fatigue of piles under different scour depths and concluded that scour has a little effect on the natural frequencies but a great effect on fatigue damage. Zhang et al. (2021) found that scour depth has a significant influence on monopile impedance. Rezaei et al. (2018) showed that scour leads to an increase in the maximum bending moment of the monopile and a shortening of the fatigue life. To mitigate the fatigue damage in OWTs, installing structural control devices is an effective way. It was demonstrated that TMDs have a positive effect on reducing vibration amplitudes of wind turbine systems (Lackner and Rotea, 2011a; Dinh and Basu, 2015; Lu et al., 2023; Aydin et al., 2023). Dai et al. (2021) conducted a shaker experiment using a scaled wind turbine model and showed that the installed TMD can suppress the vibration of the structure more effectively considering soil-structure interaction (SSI).

**Comment 2**:*L. 77: I think that the statement "This study can provide a guidance for the fatigue life evaluation […]" is exaggerating, as you use a simplified fatigue life analysis and there is other work really focusing on this topic. With the second part on TMD, I totally agree, as this is the core of your work.*

**Response:** The author agrees with your suggestion. In this study, the simplified wind turbine model is used for fatigue life analysis, which is different from a fully coupled and refined wind turbine model. According to your suggestion, the manuscript has been revised.

**Revised manuscript:**

**L89-L92:** This study provides some knowledge of the effects of the time varying scour and the TMD on the fatigue life of wind turbines, as well as a new TMD design method targeting at enhancing fatigue resistance.

**Comment 3**:*Fig. 1: In the caption of the Fig. 1, you write "scour effect" and yes, it is shown in the figure. However, it is not marked. I think it would help to mark it.*

**Response:** There is indeed a lack of markers of scour pits in Fig.1. According to your comment, the authors have improved Fig.1.

**Revised manuscript:**

[Figure]

Fig. 1. Schematic of NREL 5MW wind turbine and scour effect

**Comment 4:** *L. 87: What do you mean by "three-dimensional beam"? It is just a standard Euler-Bernoulli or Timoshenko beam?*

**Response:** Thank you for your suggestion. The "three-dimensional beam" is the three-dimensional Euler-Bernoulli beam, and the theoretical basis is still the standard Euler-Bernoulli beam theory.

**Revised manuscript:**

**L103-L104:** Three-dimensional beam elements are used to create the FE model and the theoretical basis is the standard Euler-Bernoulli beam theory.

**Comment 5:** *L. 88-91: You use only a few beam elements. Is the number sufficient? Have you conducted a convergence study? Please, show it.*

**Response:** The FE model was established in MATLAB software. The author divides the tower into 22 beam units, in which the tower is divided into 18 beam elements and the monopile between the mudline and the mean sea level (MSL) are divided into 4 beam elements. A convergence test is carried. The first natural frequency of the wind turbine is 0.2648Hz when using 22 beam elements. Then the wind turbine structure is divided into 100 elements, and it is found the corresponding first natural frequency is 0.2653Hz. The error is 0.2%, which indicates 22 beam elements are sufficient.

**Revised manuscript:**

**L104-L107:** The wind turbine tower is divided into 18 beam elements, and the monopile between the mudline and the mean sea level (MSL) are divided into 4 beam elements. A convergence test by comparing the first natural frequencies shows that 22 beam elements are sufficient.

**Comment 6:** *L. 83-94: Perhaps, a figure showing the FE model would help to see where the loads are applied, the TMD is positioned etc.*

**Response:** In Figure 4, the author shows the installation position of the TMD, which is installed inside the top of the tower barrel of the wind turbine. The wind load acts on the rotor and on the tower above the mean sea level. The wave load acts on the pile foundation at sea level and below. According to your comments, the authors add Figure 2 to show the load application and the TMD position.

**Revised manuscript:**

**L158-L159:** The application of wind and wave loads is shown in Fig. 2.

[Figure]

Fig. 2 Schematic of wind turbine load application

***Comment 7:*** *Equation 1 and 2: Are $C_T$ and $c_T$ the same (same for $K_T$ and $k_T$)?*

**Response:** The dimensions of equation 1 and 2 are ambiguous, and the authors have modified the equation 1 and 2. In equation 1 and 2, $\boldsymbol{C_T}$ and $c_T$ are not the same, and $\boldsymbol{C_T}$ is the matrix containing $c_T$. Similarly, $\boldsymbol{K_T}$ and $k_T$ are not the same, $\boldsymbol{K_T}$ is a

matrix containing $k_T$. $\mathbf{C_T} = \begin{bmatrix} 0 & \cdots & 0 \\ \vdots & \ddots & \vdots \\ 0 & \cdots & c_T \end{bmatrix}, \mathbf{K_T} = \begin{bmatrix} 0 & \cdots & 0 \\ \vdots & \ddots & \vdots \\ 0 & \cdots & k_T \end{bmatrix}, \mathbf{U_T} = \begin{bmatrix} 0 \\ \vdots \\ u_T \end{bmatrix}, \mathbf{U_s} = \begin{bmatrix} u_{s-1} \\ \vdots \\ u_{s-top} \end{bmatrix}$

**Revised manuscript:**

**L118-L130:** The TMD is mounted on the top of the tower, and the effect of the TMD is considered by adding its mass, damping, and stiffness terms at relevant positions in the local mass, damping, and stiffness matrices of the beam element representing the tower top. The equation of motion of the OWT main structure is:

$$M_s\ddot{U}_s + C_s\dot{U}_s + K_sU_s + C_T(\dot{U}_s - \dot{U}_T) + K_T(U_s - U_T) = F_{wind} + F_{wave}, \quad (1)$$

where $M_s$, $C_s$, $K_s$ are the mass, damping and stiffness matrices of the main structure. $C_T$, $K_T$ are matrices with same dimensions containing $c_T$, $k_T$. $U_s$ is the displacement vector of the main structure, and $U_T$ is the displacement vector containing $u_T$. $F_{wind}$, $F_{wave}$ are the aerodynamic and wave load vectors. The equation of motion for the TMD can be represented by

$$m_T\ddot{u}_T + c_T(\dot{u}_T - \dot{u}_{s-top}) + k_T(u_T - u_{s-top}) = 0, \quad (2)$$

where $m_T$, $c_T$, $k_T$ are the mass, damping and stiffness of the TMD, $u_T$, $u_{s-top}$ are the displacement of TMD and the displacement of the top node. The modelling of SSI is realized by an equivalent stiffness matrix, which will be introduced in detail subsequently in Section 2.3.

***Comment 8:*** *L. 102: Is $u_s$ actually the displacement vector of the tower top? Isn't it the displacement vector of all nodes of the main structure?*

**Response:** The authors have modified the equation 1 and 2. In order to make it easier for the reader to understand, the authors have changed $u_s$ to $u_{s-top}$. In Equations 1 and 2, $U_s$ is a vector containing $u_{s-top}$, where $U_s$ represents the displacement of all the nodes of the main structure, and $u_{s-top}$ represents the displacement of the node at the top of the tower of the main structure.

***Comment 9:*** *L.109-124: If I understand it correctly, you do not use a wind turbine controller when calculating the wind load from the turbulent wind field. This is a significant simplification. I do not know how important this simplification is in this context, but it might be relevant. At least, you have to discuss this simplification.*

**Response:** The effect of controller was considered in a simplified way in this article.

The presumed relationships between the wind speed, rotor rotation speed and pitch angle are used to capture the effect of pitch control and generator torque control. A sentence has been added in the manuscript to clarify this point:

**Revised manuscript:**

**L148-151:** To represent the influence of controller in the OWT, a standard relationship (J. Jonkman et al., 2009) between the mean wind speed, rotor rotation speed and blade pitch angles, which represents the OWT's normal operational conditions, are adopted throughout the wind loading calculation.

*Comment 10*:*Section 2.3: You discuss that you use the more complex ABAQUS model and not simplified p-y curves for the derivation of your stiffness matrix. This is totally fine. However, the soil stiffness is load dependent. The load dependency is even represented by the p-y curves, but not by your stiffness matrix. You definitely have to discuss the load dependence of the soil stiffness.*

**Response:** As you said, the soil stiffness is load dependent. In this study, the authors use a complex ABAQUS model to better simulate soil. The correlation between soil stiffness and load is not shown in the equivalent stiffness matrix, because the nonlinearity between soil stiffness and load for the wind turbine operational condition is very weak in the equivalent stiffness matrix. The authors ignored the nonlinearity during the study and only performed the linearization equivalent. In the equivalent stiffness matrix, the torsional stiffness plays a major role. For example, when the scour depth is 1.4 times the pile diameter, the torsional stiffness is almost linear. The torsional stiffness is shown in the figure below, and the load dependence of the equivalent stiffness matrix is small.

[Figure]

Linear equivalent diagram of torsional stiffness

**Comment 11:** *Eq. 6: I think it should be $\boldsymbol{\Psi^T M \Psi \ddot{a} + \Psi^T C \Psi \dot{a} + \Psi^T K \Psi a = \Psi^T F}$ and not $\boldsymbol{\Psi^T M \Psi \ddot{u} + \Psi^T C \Psi \dot{u} + \Psi^T K \Psi u = \Psi^T F}$ as the transformation is $\boldsymbol{u = \Psi a}$.*

**Response:** Thank you for your suggestion. As you said, according to the transition equation $u = \Psi \alpha$, the equation 6 should be $\Psi^T M \Psi \ddot{a} + \Psi^T C \Psi \dot{a} + \Psi^T K \Psi a = \Psi^T F$ According to your comment, the authors have revised the manuscript.

**Revised manuscript:**

**L257-L259:** According to relationship $\boldsymbol{u = \Psi \alpha}$ and multiplying the transpose of the undamped vibration matrix $\boldsymbol{\Psi^T}$ with the equation of motion, the following equation is obtained:

$$\boldsymbol{\Psi^T M \Psi \ddot{a} + \Psi^T C \Psi \dot{a} + \Psi^T K \Psi \alpha = \Psi^T F}. \tag{6}$$

**Comment 12:** *Eq. 8 is not sufficiently clear. For example, x and y are not explained. Furthermore, the element shape functions are neither given nor explained. It is not stated that the shape functions refer to the original FE model, i.e., the model before applying the modal reduction.*

**Response:** According to your comments, the authors make a more detailed explanation about equation 8 and have modified the manuscript to make it easier for readers to

understand. The authors have added coordinate axes in Figure 2 for easy understanding.

**Revised manuscript:**

**L271-L273:** where $\mathbf{u}^e$ is the nodal displacement vector at the cross section, E is the material elastic modulus, and $\mathbf{N}^e$ is the elemental shape function vector of FE model, x and y are the positions within the section at the height z of the tower.

***Comment 13:*** *L. 243: "t is the thickness at which cracks may grow"; are you sure that this statement is correct? Isn't t the actual thickness of the pile?*

**Response:** According to the description of Equation 9 in reference paper "RP-C203: Fatigue design of offshore steel structures", t is the thickness through which a crack will most likely grow. And $t = t_{ref}$ is used for thickness less than $t_{ref}$. In fact, when t is more than $t_{ref}$, t is the actual thickness of the pile.

References

DNVGL-RP-0005. (2014a). RP-C203: Fatigue design of offshore steel structures.

***Comment 14:*** *Table 3: A SCF is given. However, you never state how you use it.*

**Response:** The author finds that the description and usage of SCF is lacking. SCF is short for "Stress concentration factor". In Equation 9, $\Delta\sigma$ is the stress range calculated from the nominal stress $\Delta\sigma_{nominal}$ by the equation $\Delta\sigma = SCF \cdot \Delta\sigma_{nominal}$. The authors have revised the manuscript to add the description and usage of SCF.

**Revised manuscript:**

**L278-L280:** where N is the number of cycles to failure, $\Delta\sigma$ is the stress range. $\Delta\sigma$ is calculated from the nominal stress $\Delta\sigma_{nominal}$ by the equation $\Delta\sigma = SCF \cdot \Delta\sigma_{nominal}$, SCF is the stress concentration factor.

**Comment 15:** *L. 251: Nc is obtained by the rainflow counting? I think Nc has to be defined before the rainflow counting can start, as it is the number of bins, the rainflow counting sorts the cycles into. Which value do you use for Nc?*

**Response:** $N_c$ in Equation 10 is the total number of bins. As you said, it is theoretically necessary to define $N_c$ first, then determine the stress range $i$ according to $N_c$, and arrange the cycles within it through rain flow. However, MATLAB has its own rainflow counting function. When the stress time history is given, the function automatically obtains the stress range $i$ and the corresponding cycle number $n_i$ according to the stress time history, and $N_c$ is the total number of stress range $i$.

**Comment 16:** *L. 272-274: I would be careful when stating that the fore-aft mode is the most important one. For large monopile and significant wind-wave-misalignments, side-to-side modes can be more critical with respect to fatigue, as the aerodynamic damping is lower in side-to-side direction.*

**Response:** For a normally operating OWT, the proportion of its power production time can reach up to more than 95% of its service life, and during the production time the tower mainly vibrates in the fore-aft direction due to variations in thrust under high aerodynamic damping from the wind-rotor interaction. It is true that when the wind turbine is parked with lower aerodynamic damping the wind-wave-misalignments can lead to significant side-side vibrations which may also cause large fatigue damage. These vibrations both belong to vibrations in the first bending mode. Relevant text has been slightly modified as below.

Furthermore, the article mainly deals with the influence of scour and installed TMD, and conducting a comprehensive fatigue analysis by covering all operating conditions is out of the scope. The authors believe that the current analysis focusing on fore-aft vibration is enough to draw main conclusions presented in this paper.

**L310-L312:** As the dominant vibration mode of the OWT structure in operation is the first bending mode  the largest vibration amplitude occurs at the tower top and installing the TMD at the tower top is most effective.

**Comment 17:** *L. 275-276: If the TMD is in the tower, is it still rotating, when the RNA is yawed? Otherwise, I wonder how the TMD can always be aligned with the fore-aft direction.*

**Response:** Your comment is valuable. The authors choose to install the TMD inside the tower barrel because the spare space of the nacelle is limited and it cannot accommodate the installation of the TMD. If there is enough free space in the nacelle, it is the best choice to install in the nacelle, which can synchronize the control direction with the RNA deflection, and ensure that the TMD control direction is aligned with the FA directions. The TMD is installed inside the tower barrel and can be aligned with the FA direction by rotation. After the RNA is yawed, it is possible to make the TMD rotate inside the tower barrel by using a rotating disc mechanism to ensure that the control direction is always aligned with the fa direction of the wind turbine. Another method is to install two TMDs responsible for vibration control in different directions.

**Comment 18:** *L. 290: Why did you choose 1% for the mass ratio and not any other value?*

**Response:** According to a large number of engineering practices, for tall building structures, the mass ratio of 1%-2% can effectively suppress the wind-induced vibration of the structure. In addition, it was found in previous studies that both Colwell and Lackner could effectively suppress vibration by using TLCD with a mass ratio of 1% and TMD with a mass ratio of 2% (Colwell and Basu, 2009; Lackner and Rotea, 2011b; R. Zhang et al., 2019). In the study, the author considers that excessive mass will lead to increased construction cost and difficulty and excessive change of the inherent

characteristics of the original structure, so the mass ratio of TMD is 1%.

**reference**

Colwell, S. and Basu, B. (2009). Tuned liquid column dampers in offshore wind turbines for structural control. Engineering Structures, 31(2), 358–368. https://doi.org/10.1016/j.engstruct.2008.09.001

Lackner, M. A. and Rotea, M. A. (2011b). Structural control of floating wind turbines. Mechatronics, 21(4), 704–719. https://doi.org/10.1016/j.mechatronics.2010.11.007

Zhang, R., Zhao, Z. and Dai, K. (2019). Seismic response mitigation of a wind turbine tower using a tuned parallel inerter mass system. Engineering Structures, 180, 29–39. https://doi.org/10.1016/j.engstruct.2018.11.020

***Comment 19:*** *Section 3.2: Perhaps, it would help to give a short example demonstrating how $c_T$ and $k_T$ change if the eigenfrequency drops to, for example, 0.26 Hz due to scour.*

**Response:** The purpose of this study is to obtain the optimal TMD parameters based on the fatigue life of wind turbines. The TMD optimization idea adopted is as follows: First, given the initial TMD parameter $m_T, c_T, k_T$, the time-varying scour depth curve is divided into different scour depths by 0.1D, and the fatigue damage of the wind turbine in each 0.1D interval is calculated. The scour depth was gradually increased by 0.1D until the fatigue damage reached unit 1, so as to obtain its corresponding fatigue life. During this process, the parameters of TMD remain unchanged. Finally, the optimization function GlobalSearch is used to change the parameter $c_T, k_T$ of TMD, and the above process is repeated to obtain a new fatigue life. The optimal TMD parameter $c_T, k_T$ corresponding to the maximum fatigue life is obtained through continuous optimization calculation. The damping and stiffness of TMD at different scour depths are shown in the figure below.

[Figure]

Damping and stiffness of TMD under different scour depth

**Comment20:** *Figure 6: Is the equivalent stiffness matrix actually "added" to the 4DOF model? I thought that it is added to the MATLAB FE model.*

**Response:** Thank you for your suggestion. In Figure 6, there is an error in the description about the position of equivalent stiffness matrix addition. As you said, the equivalent stiffness matrix is added to the MATLAB FE model, and then the 4DOF modal model is obtained through modal decomposition. The authors have revised the original manuscript.

**Revised manuscript:**

[Figure]

Fig. 6. Flowchart of TMD fatigue-life-based optimization technique

**Comment 21:** *Figure 6: The parts on "Divided by 0.1D" and "Scour depth plus 0.1D" are completely unclear at this stage. They become a bit clearer later on, but I think some explanation or at least a reference to a later section is needed here. Otherwise, the reader is lost.*

**Response:** The authors find that the description of the TMD optimization process in Figure 6 is lacking. According to your comments, the authors have revised the original manuscript to add a description of "Divided by 0.1D" and "Scour depth plus 0.1D".

**Revised manuscript:**

**L348-L356:** In this technique, the frequency ratio, mass ratio and damping ratio of the TMD are set as the optimal parameters to search, and the fatigue life is the optimization objective. When considering the time-varying scour process, the time-varying scour depth curve is first divided into a number of scour depths with an increment of 0.1d.

For each scour depth, the fatigue damage is calculated respectively and then the total fatigue damage in a particular duration can be summarised. When the scour pit becomes deeper, the fatigue damage accumulates and finally reaches unit 1 which denotes the end of fatigue life. The simplified 4-DOF modal model incorporating scour modelling is used to generate the stress time series.

*Comment 22*:*L. 309: Here, you state that the mass ratio is a variable. Before, you just select a value, i.e., 1%, for it. Later, you do both. This is quite confusing when you read the paper for the first time. Perhaps, it would help to elaborate a bit more on it it (see comment 17 as well)*

**Response:** In the research process, the author first set the mass ratio to 1%, and also set the mass ratio of TMD to 1% when optimizing TMD parameters. The author only optimized the parameter frequency ratio and damping ratio, and compared the fatigue life of wind turbine after TMD optimization. Subsequently, in order to understand the fatigue life of wind turbines after TMD optimization when the value of TMD mass ratio is not fixed, the author gives a mass ratio optimization interval, making the mass ratio a variable within the optimization interval. According to your comments, the authors have revised the original manuscript.

**Revised manuscript:**

**L359-L363:** In the TMD optimization process, the mass ratio of TMD is first set to 1%, and only the parameter frequency ratio and damping ratio are optimized. Subsequently, in order to understand the optimization effect of TMD when the value of TMD mass ratio is not fixed, a mass ratio optimization interval is given, so the mass ratio becomes a variable within the optimization interval.

*Comment 23*:*L. 313: You state that you use Fmincon. First of all, Fmincon is just a MATLAB routine. What is actually interesting is which optimization algorithm is used.*

*If I remember correctly, Fmincon uses a local optimization algorithm. Is this sufficient? Have you looked at the objective space and it is rather smooth without local minima? Otherwise, a global optimization algorithm might be more appropriate.*

Response: Thank you for your suggestion. According to your comments, the authors use Fmincon function and GlobalSearch function respectively to optimize the TMD parameters based on fatigue damage. The Fmincon function is a local optimization function with high optimization efficiency, and the GlobalSearch function is a global optimization function with large amount of calculation and lower optimization efficiency. In order to compare the optimization results of Fmincon function and GlobalSearch function, the author reduced the load case reasonably and carried out the optimization in the same case using Fmincon function and GlobalSearch function respectively. The frequency ratio of the TMD is 0.9291, the damping ratio is 0.0467, and the fatigue life is 69 years, as obtained from the Fmincon function. And the frequency ratio of the TMD is 0.8708, the damping ratio is 0.0132, and the fatigue life is 73.8 years, as obtained from the GlobalSearch function. The fatigue life error is 7% obtained by optimization using Fmincon function and GlobalSearch function.

Therefore, the authors choose to use the GlobalSearch function instead of Fmincon for optimization. Finally, the TMD frequency is 0.9432, the damping ratio is 0.0496, and the fatigue life is 93.18 years.

**Revised manuscript:**

Table 6. Optimization of TMD parameters

| Optimization method | Mass ratio range | Time-varying scour | Optimal mass ratio | Optimal frequency ratio | Optimal damping ratio | Fatigue life (Year) |
|---|---|---|---|---|---|---|
| Initial (LC 5) | 0.01 | Use | 0.01 | 0.99 | 0.061 | 74.6 |
| FOT | 0.01 | Use | 0.01 | 0.94 | 0.050 | 93.2 |
| FOT | 0.001-0.1 | Use | 0.097 | 0.92 | 0.15 | 133.2 |

**L540-L548:** It shows that when the mass ratio is fixed at 1%, the optimal frequency ratio is 0.94, the optimal damping ratio is 5%, and the final fatigue life is 93.2 years. Compared to the fatigue life with initially optimized TMD using the traditional method without considering scour, the fatigue life is increased by 18.6 years or about 25%. It indicates that the parameter search in the optimization process is correct and it is optimal to use the TMD parameter search method to design the TMD after obtaining the time-varying scour curve. When the mass ratio range is taken from 0.1% to 10%, the optimal mass ratio of the TMD is 9.7%, the frequency ratio is 0.92, the damping ratio is 15%, and the final fatigue life is 133.2 years.

***Comment 24:*** *L. 318: You state that you model operational and parked conditions. What is the difference between these two in your simplified model without a controller? Are only the wind loads different or do you also change the inertia of the RNA etc.? In reality, even the first fore-aft bending eigenfrequencies of the entire turbine are slightly different in operational and parked conditions.*

Response: When the author simulates the operational and parked conditions of the wind turbine, the difference is in the wind loads. In the operating conditions, the author considers the aerodynamic load on the rotating rotor and the wind load on the tower. In this case, the wind load on the rotor is calculated by blade element momentum theory. When the ambient wind speed is lower than the cut-in wind speed or higher than the cut-out wind speed, the wind turbine parks, and the blade pitch Angle is 90 degrees. At this time, the wind turbine mainly bears the aerodynamic load on the tower, and the aerodynamic damping is very small. The aerodynamic loading on the blades is calculated by directly looking at the aerodynamic loading coefficient table given the local attack angles. In this case, the total aerodynamic loading on the rotor is much smaller. In this study, the authors do not consider the changes in dynamic characteristics under the operational and parked conditions.

**Comment 25:** *Table* 4: $V_w, T_z, H_s$ *and* $P_{state}$ *are not explained.*

**Response:** According to your comments, the authors have modified the original manuscript to add the explanations of $V_w, T_z, H_s$ and $P_{state}$.

**Revised manuscript:**

**L382-L383:** In Table 4, $V_w$ is the wind speed, $T_z$ is the zero-crossing wave period, $H_s$ is the wave height, and $P_{state}$ is the probability of environmental state.

**Comment 26:** *L. 317-319: You state that you have 22 environmental states for operational and parked situations. However, only in line 401, you start to explain that you run six 10-min simulations for each condition. This should already be stated here. Furthermore, two questions are unanswered in my opinion: 1) What is the total number of simulations? Is it 2 × 22 × 6 (operating/parked x environmental states x seeds)? 2) Do you remove some time at the beginning of each simulation to remove initial transients? If yes, how much?*

**Response:** The authors agree with you very much and have modified the original manuscript to show the description of simulation time for each working condition in front of Table 4.

Question 1: As you said, the authors simulated operational and parked states with six seeds, 22 environmental states, and 14 different scour depths. The scour depth is divided into 0.1D and simulated for 1.3D. The total number of simulations is $2 \times 6 \times 22 \times 14$.

Question 2: The authors simulate a total of 700 seconds and remove 100 seconds of the initial transient at the end of each simulation, retaining 600 seconds of simulated data.

**Revised manuscript:**

**L375-L380:** For a particular set of mean wind speed, wave period and wave height, six different random seed numbers are used to generate different wind fields and wave

profiles to reduce the influence of randomness. To obtain the stress time histories at the mudline, a 700s simulation for each random seed was conducted and the response in the first 100 seconds was deducted to eliminate the effect of initial transient vibration.

***Comment27:*** *L. 338-339: What are the values for d50, $\rho_s$ and $\rho_w$ you use?*

**Response:** Thank you for your suggestion. According to your comments, the authors have modified the original manuscript to add the values of $d_{50}, \rho_s$ and $\rho_w$.

**Revised manuscript:**

**L394-L397:** u is the tidal velocity and taken as 0.5 m/s, $u_c$ is the critical shear velocity and taken as 0.37 m/s, g is the acceleration of gravity and taken as 9.8 m/s$^2$, $d_{50}$ is grain size of sea sand and taken as 0.2 mm. The parameter $\Delta = \frac{\rho_s}{\rho_w} - 1$, where $\rho_s$ is density of sand and taken as 2.65 g/cm$^3$, $\rho_w$ is density of water and taken as 1 g/cm$^3$.

***Comment 28:*** *Fig. 6: Perhaps, it would be nice if you add two other graphs to this figure for the TMD (ABAQUS + MATLAB) or at least one for MATLAB if you do not have the TMD implemented in ABAQUS.*

**Response:** According to your comments, the authors have modified the original manuscript. In Figure 6, the authors add the relationship curve between frequency and depth of MATLAB model including TMD.

**Revised manuscript:**

[Figure]

Fig. 8. Relationship between wind turbine natural frequency and scour depth

**Comment 29**:*L. 371: Is this case an operating or a parked case?*

**Response:** The author does not state clearly the status of the wind turbine. In the ninth environmental state, the author calculates the wind turbine in the operating state and compares the response of the tower top displacement. The authors have modified the original manuscript.

**Revised manuscript:**

**L436-L438:** When the OWT in the operating state is under the 9th environmental state which corresponds to the rated wind speed of 12 m/s, a comparison for the tower top displacements is made for LC 1, LC 3, LC 4 and LC 6, as shown in Fig. 10.

**Comment 30**:*L. 379: You state that the effect is more prominent for other operating conditions. First, I think that is it especially more prominent for parked conditions with less aerodynamic damping. And second, please show a case, where the effect is more prominent. You can just add a second figure.*

**Response:** Thank you for your suggestion, The author fully agrees with you. The statement about "The effect of the TMD is more prominent for other operating conditions with less aerodynamic damping." is inaccurate. What the author wants to express is the same as you think, "the effect is more prominent for parked conditions with less aerodynamic damping." The author reformulates the statement and presents a case based on your comments. The authors compared the tower top displacement response when the wind turbine is installed with and without TMD under the parked condition with the 3m/s wind speed. The authors have modified the original manuscript.

**Revised manuscript:**

**L442-L447:** It is known that the aerodynamic damping is large when the OWT is operating under the rated wind speed, so it is normal that the vibration mitigation effect of the TMD is less significant in this case. The effect of the TMD is more prominent for parked conditions with less aerodynamic damping. As shown in the Fig. , the vibration mitigation effect of the TMD is more significant under the parked condition with 3 m/s wind speed.

[Figure]

Fig. 11 The displacement response of wind turbine tower under the parked condition with 3 m/s wind speed

*Comment 31*:*Fig. 7: I can hardly read this figure in greyscale. Please, enlarge it and think about clearly different line styles (and perhaps also thicker lines).*

**Response:** The authors have changed Figure 7 to a vector diagram and enlarged it further. Now the curve in the figure can be clearly seen.

*Comment 32*:*Fig. 7: In the caption, you write "four operating conditions". I think it should be "four load cases".*

**Response:** Regarding the comparison of different curves in Figure 7, the authors use the same environmental state, so the wind turbine is subjected to the same load. The difference in Figure 7 is that the authors performer different operating states,

respectively installed and not installed TMD, and the scour depth is 0D and 1.3D. Therefore, the author thinks it is better to use "four operating conditions".

**Comment 33:** *Section 4.4: You visually compare time series and spectra for the 4DOF and the FE model. This is a good starting point. However, frequently, you cannot see the differences leading to different fatigue lifetimes in these plots immediately. Hence, it would be good if you could also calculate the damage value Dk (Eq. 10) for this time series and the two models. This would be an objective comparison.*

Response: According to your comments, the authors calculate the fatigue damage caused by the stress time history curve of the FE model and 4-DOF model in Figure 10. The fatigue damage caused by FE model in 10 min is $2.108 \times 10^{-7}$, and the fatigue damage caused by 4DOF model in 10 min is $2.1 \times 10^{-7}$, with an error of 0.05%. The authors have modified the original manuscript to show $D_k$ values and errors.

**Revised manuscript:**

**L465-L467:** The fatigue damage caused by the FE model in 10 min is $2.108 \times 10^{-7}$, and the fatigue damage caused by the 4-DOF model in 10 min is $2.1 \times 10^{-7}$, with an error of 0.05%.

**Comment 34:** *L. 434 and 445: How does these two values (50% and 62%) fit together?*

**Response:** Thank you for your suggestion. In Figure 9, when no scour occurs, the fatigue life of the wind turbine with TMD is 90 years, and the fatigue life of the wind turbine without TMD is 59.3 years, and the comparison shows that the fatigue life of the wind turbine increases by about 51.8% after the installation of TMD. Similarly, compared with LC1 and LC4 in Figure 10, the fatigue life of the wind turbine increased by about 30.7 years, or about 51.8%, after the installation of TMD.

**Revised manuscript:**

**L513-L516:** When comparing the results for LC 1 and LC 4, it shows the installation of the TMD results in a significant increase in the fatigue life of the OWT, with an increase in fatigue life of about 30.7 years, which is about 51.8%.

*Comment 35:You use your "standard" value of 1% as a boundary value for the optimization. This is not a good approach, as it excludes all values below the "standard" value. Please, either repeat your optimization with another boundary value or justify your choice.*

Response: According to your comments, the authors select 0.1% as the lower boundary value of the optimized interval. The authors have modified the original manuscript.

**Revised manuscript:**

**L531-L534:** A range of the mass ratio from 0.001 to 0.1 is used to optimize the TMD so that the influence of the mass ratio can be evaluated.

Table 6. Optimization of TMD parameters

| Optimization method | Mass ratio range | Time-varying scour | Optimal mass ratio | Optimal frequency ratio | Optimal damping ratio | Fatigue life (Year) |
|---|---|---|---|---|---|---|
| Initial (LC 5) | 0.01 | Use | 0.01 | 0.99 | 0.061 | 74.6 |
| FOT | 0.01 | Use | 0.01 | 0.94 | 0.050 | 93.2 |
| FOT | 0.001-0.1 | Use | 0.097 | 0.92 | 0.15 | 133.2 |

*Comment 36:Table 6: I think that it would help if you name the first row "Initial (LC 5)". This would make things much clearer.*

Response: Thank you for your suggestion. According to your comments, the authors have modified the original manuscript.

**Revised manuscript:**

Table 6. Optimization of TMD parameters

| Optimization method | Mass ratio range | Time-varying scour | Optimal mass ratio | Optimal frequency ratio | Optimal damping ratio | Fatigue life (Year) |
|---|---|---|---|---|---|---|
| Initial (LC 5) | 0.01 | Use | 0.01 | 0.99 | 0.061 | 74.6 |
| FOT | 0.01 | Use | 0.01 | 0.94 | 0.050 | 93.2 |
| FOT | 0.001-0.1 | Use | 0.097 | 0.92 | 0.15 | 133.2 |

**Comment 37:** *L. 471: You state that your results indicate that "considering time-varying scour depth" is beneficial. However, you cannot know this from your results, as you directly compare "fixed" TMD parameters with optimized ones which consider time-varying scour depth. What you do not compare are optimized TMD parameters for the maximum scour depth. I can imagine that these are quite similar to the ones you have determined for the time-varying scour depth. Hence, perhaps, the benefit is just due to the optimization. Therefore, you should either include TMD parameters that are optimized for the maximum scour depth in your analysis, or you should it least discuss this aspect here.*

Response: Thank you for your suggestion. In fact, when GlobalSearch function optimization is adopted, the parameter screening process of TMD includes not only the TMD parameters obtained by the traditional TMD design method, but also the TMD parameter conditions designed according to the maximum scour. Compared to the fatigue life with the initially optimized TMD using the traditional method without considering scour, the fatigue life is increased by 18.6 years or about 25 percent. Further, by comparing the results for LC 5 and LC 6, we can know that with the same initial designed TMD installed considering maximum scour depth or the time-varying scour does not have large impact on the fatigue life (74.1 years vs 74.6 years). This observation implies that a better fatigue resistance can be obtained only by combing time-varying scour and corresponding TMD design method. Also, it means that the parameter search in the optimization process is correct. Therefore, it is better to use the

TMD parameter search method to design the TMD given the time-varying scour curve. The authors have modified the original manuscript.

**Revised manuscript:**

**L541-L543:** It indicates that the parameter search in the optimization process is correct and it is better to use the TMD parameter search method to design the TMD after obtaining the time-varying scour curve.

*Comment 38*:*Conclusions: You should clearly state your major simplifications, e.g., simplified lifetime calculation with just 22 environmental states, no controller, TMD only in fore-aft direction etc.*

Response: The authors very much agree with you that the simplification of the model is not stated clearly enough in the conclusion. The authors have modified the original manuscript to add a simplified description of the model, including the adopted environment, the simplification of the controller and the working direction of TMD etc.

**Revised manuscript:**

**L550-L557:** This study establishes a rapid numerical model which can consider the effect of scour and installation of a TMD, in which the TMD operates only in the FA direction. The model is simplified by using concentrated mass instead of RNA and ignores the nonlinearity of the equivalent stiffness matrix. The established model is used to investigate the influence of scour and the installed passive structural control device on the OWT's natural frequencies and fatigue life through 22 environmental states. An optimization technique is also developed to find optimal parameters of the TMD considering time-varying scour. It shows that the vibration amplitude of the OWT can be effectively reduced by the TMD.

Typos etc.:

1) As you can see in the following, there are some typos and inconsistencies. As I have definitely not found all of them, I recommend a thorough proof reading.

2) Please, check your citation style, e.g., in line 43, it should be "Sørensen and Ibsen, 2013". The other names are given names.

3) L. 49 and others: "damage" and not "damages". There is no plural of "damage" in the context of structural engineering.

4) L. 83: "An FE model" and not "A FE model"

5) L. 83: "a monopile-supported OWT" and not "an monopile-supported OWT"

6) Table 1: "Rated wind speed" not "Rated wind Speed"

7) L. 129: "in Shirzadeh et al. (2013)" and not "in Ref. (Shirzadeh et al., 2013)"

8) Table 2: "kN/m³" and not "kN/m3"

9) L. 227: Remove "the" before "Eq. (6)

10) L. 287 "and $\xi opt$ is the" not "and is the"

11) L. 296: "885 $Ns/m$" and not "885 $N \cdot s/m$"

12) L. 333: Remove "was used"

13) L. 406: I think it should be "where the maximum stress is reached".

14) L. 484: Where is the "on the one hand"? You just use "on the other hand.

15) L. 578: "Patil" not "patil".

16) L. 594: See typo comment 2.

17) L. 628 "van der Tempel, J. (2006)" and not „Tempel, J. van der. (2006)"

**Response:** Thank you for your suggestion. The author has found that there are some problems with typos and inconsistencies. Thank you very much for your help in finding out various problems. According to your comments, the authors have revised them one by one and further proofread the full text to avoid similar problems.

---

## Referee Report (RR1)

**"Effect of scour on the fatigue life of offshore wind turbines and its prevention through passive structural control" (Manuscript number: wes-2023-149 – first revision)**

Thank you very much for the comprehensive revision of the manuscript. My questions have been answered completely. I only think that some more of your answers should be added to the manuscript. Therefore, I have a few last suggestions, which should be considered in a final minor revision.

Points to be added to the manuscript:

1) Previous comment no. 7: Could you please add the matrices $C_T = \begin{bmatrix} 0 & \cdots & 0 \\ \vdots & \ddots & \vdots \\ 0 & \cdots & c_T \end{bmatrix}$ etc. to the manuscript. In your answer, the definition of $C_T$ it is very, but in the manuscript, it can still be improved.

2) Previous comment no. 10: In your answers, you nicely explain that and why you neglect the nonlinearity of the soil matrix. In the paper, it is only mentioned in the conclusions. Perhaps, you can add one more sentence to Section 2.3.

3) Previous comment no. 13: According to the reference paper "t is the thickness through which a crack will most likely grow. And t =tref is used for thickness less than tref. In fact, when t is more than tref, t is the actual thickness of the pile." In your case, t>tref. Hence, t is the actual thickness of the pile. I suggest adding this to the manuscript.

4) Previous comment no. 15: In your answers, you explain that $N_C$ is obtained by the rainflow counting algorithm in MATLAB. As the reader might not know this, you should add an explanation how $N_C$ is obtained to the manuscript.

5) Previous comment no. 17: You comprehensively explain the reason for putting the TMD in the tower. I think that this explanation is not required to be repeated in the manuscript. Nonetheless, in the manuscript, it should be pointed out that the TMD could/should be rotatable as well (second last sentence of your explanation in the answers).

6) Previous comment no. 24: You give a description of the difference between the modelling of operating and parked conditions in the answers. However, none of the explanation is added to the manuscript. Perhaps, you add the most important points of it to the manuscript as well.

---

## Author Response (AR2)

**Reply to Reviewers' and Editor's Comments**

**(Manuscript number: wes-2023-149)**

**Effect of scour on the fatigue life of offshore wind turbines and its prevention through passive structural control**

**By Yu Cao, Ningyu Wu, Jigang Yang, Chao Chen, Ronghua Zhu, Xugang Hua**

**Overall Response:**

We are very grateful to the Editor and Reviewers for their constructive comments on this manuscript once again. We have further revised and improved the manuscript based on the comments and clarified the issues brought up in the paper. In the following sections, point-by-point responses to the comments were provided. The original comments are in italics. The authors' responses are highlighted in blue. The corresponding changes are highlighted in red in the revised manuscript.

**Reviewer #2**

***Comment 1:*** *Previous comment no. 7: Could you please add the matrices $C_T = \begin{bmatrix} 0 & \cdots & 0 \\ \vdots & \ddots & \vdots \\ 0 & \cdots & c_T \end{bmatrix}$ etc. to the manuscript. In your answer, the definition of $C_T$ it is very, but in the manuscript, it can still be improved.*

**Response:** Thank you for your suggestion. According to your comments, the authors have further added the formulations of the matrices $C_T, K_T, U_S$ and vector $U_T$ in the original manuscript.

**Revised manuscript:**

**L122-L126:** where $\mathbf{M}_s, \mathbf{C}_s, \mathbf{K}_s$ are the mass, damping and stiffness matrices of the main structure. $\mathbf{C}_T, \mathbf{K}_T$ are matrices with same dimensions containing $c_T, k_T$ , $\mathbf{C}_T =$

$$\begin{bmatrix} 0 & \cdots & 0 \\ \vdots & \ddots & \vdots \\ 0 & \cdots & c_T \end{bmatrix}, \mathbf{K}_T = \begin{bmatrix} 0 & \cdots & 0 \\ \vdots & \ddots & \vdots \\ 0 & \cdots & k_T \end{bmatrix}.$$ $\mathbf{U}_s$ is the displacement vector of the main structure,

$$\mathbf{U}_s = \begin{bmatrix} u_{s-1} \\ \vdots \\ u_{s-top} \end{bmatrix}.$$ $\mathbf{U}_T$ is the displacement vector containing $u_T$, $\mathbf{U}_T = \begin{bmatrix} 0 \\ \vdots \\ u_T \end{bmatrix}.$ $\mathbf{F}_{wind}$, $\mathbf{F}_{wave}$

are the aerodynamic and wave load vectors.

***Comment 2:*** *Previous comment no. 10: In your answers, you nicely explain that and why you neglect the nonlinearity of the soil matrix. In the paper, it is only mentioned in the conclusions. Perhaps, you can add one more sentence to Section 2.3.?*

**Response:** According to your comments, the authors have added explanations about ignoring nonlinearity of the soil matrix in Section 2.3.

**Revised manuscript:**

**L232-L236:** where $\mathbf{K}_{soil}$ is the equivalent soil stiffness matrix, $\mathbf{u}_{soil}$ is the displacement vector, and $\mathbf{F}_{soil}$ is the reaction force vector. The equivalent soil stiffness matrix ignores the nonlinearity in the force-displacement relationship. This approach is suitable for fatigue analysis, as in normal operation conditions the deformation of the soil around the monopile is relatively small and the nonlinearity in soil stiffness is very weak.

***Comment 3:*** *Previous comment no. 13: According to the reference paper "$t$ is the thickness through which a crack will most likely grow. And $t = t_{ref}$ is used for thickness less than $t_{ref}$. In fact, when $t$ is more than $t_{ref}$, $t$ is the actual thickness of the pile." In your case, $t > t_{ref}$. Hence, $t$ is the actual thickness of the pile. I suggest adding this to the manuscript.*

**Response:** According to your comments, the authors have added more details of the value of $t$ in the manuscript.

**Revised manuscript:**

**L282-L288:** where $N$ is the number of cycles to failure, $\Delta\sigma$ is the stress range. $\Delta\sigma$ is calculated from the nominal stress $\Delta\sigma_{nominal}$ by the equation $\Delta\sigma = SCF \cdot \Delta\sigma_{nominal}$, SCF is the stress concentration factor. $m$ is the negative inverse slope of the S-N curve, and $log\bar{a}$ is the intercept between the log N axis and the S-N curve, $t_{ref}$ is the reference thickness for welded joints, $t$ is the thickness at which cracks may grow. A*nd $t = t_{ref}$ is used for thickness less than $t_{ref}$. When $t$ is larger than $t_{ref}$, $t$ is the actual thickness of the pile.* $k$ is the thickness exponent of fatigue strength.

***Comment 4:*** *Previous comment no. 15: In your answers, you explain that $N_c$ is obtained by the rainflow counting algorithm in MATLAB. As the reader might not know this, you should add an explanation how $N_c$ is obtained to the manuscript.*

**Response:** According to your comments, the authors have added an explanation for how $N_c$ is obtained.

**Revised manuscript:**

**L295-L300**: where $\mathbf{N_c}$ is the total number of bins obtained by rainflow counting, $\mathbf{n_i}$ is the number of cycles in stress range $\mathbf{i}$, $\mathbf{N_i}$ is the number of cycles to failure in stress range $\mathbf{i}$ , and $\mathbf{D_k}$ is the total fatigue damage index. The "rainflow" function in MATLAB is adopted for rainflow counting. When a stress time history is given, this function can automatically obtain the $\text{i}^{th}$ stress range and the corresponding cycle number $n_i$, and $N_c$ is the total number of stress ranges.

***Comment 5:*** *Previous comment no. 17: You comprehensively explain the reason for putting the TMD in the tower. I think that this explanation is not required to be repeated in the manuscript. Nonetheless, in the manuscript, it should be pointed out that the TMD could/should be rotatable as well (second last sentence of your explanation in the answers).*

**Response:** Thank you for your suggestion. According to your comments, the authors have added an instruction about the TMD can be rotated in the tower to the manuscript.

**Revised manuscript:**

**L321-L323:** Therefore, the TMD is installed inside the steel tube at the tower top to mainly control the vibration in the FA direction, as shown in Fig. 5. And the TMD is aligned with the FA direction by rotating the damper.

*Comment 6:Previous comment no. 24: You give a description of the difference between the modelling of operating and parked conditions in the answers. However, none of the explanation is added to the manuscript. Perhaps, you add the most important points of it to the manuscript as well*

**Response:** Thank you for your suggestion. According to your comments, the authors have added an explanation of the difference between the modelling of operating and parked conditions to the manuscript.

**Revised manuscript:**

**L379-L386:** In this study, fatigue analyses are performed under 22 environmental states provided by Tempel (2006), taking into account both operational and parked conditions. These 22 environmental states are shown in Table 4. In operating conditions, the wind turbine bears the aerodynamic load of the rotating rotor and the wind load of the tower, and the wind load on the rotor is calculated using the BEM theory. In parked conditions, the wind turbine mainly bears the aerodynamic load on the tower, and the aerodynamic damping is very small. The aerodynamic loading on the blades is calculated by directly looking at the aerodynamic loading coefficient table given the local attack angles.